# ImpuGen: Unified Diffusion-based Tabular Imputation and Generation via Task-Aligned Sampling Strategies

## Abstract

Imputation of missing values and tabular data synthesis both rely on distribution modeling, but they pursue different goals. pointwise accuracy is required in imputation, whereas diversity and fidelity are crucial in generation. We present ImpuGen, a single conditional diffusion model that achieves both objectives. ImpuGen employs two efficient task-aligned sampling strategies. (i) A zero-start sampling, which yields accurate, deterministic imputations without multiple-sample averaging. (ii) A distribution-matching refinement (DMR), which randomly remasks columns with probability $p$ and regenerates them to reduce distributional mismatch. Across nine public datasets, ImpuGen surpasses eleven imputation baselines—reducing MAE by up to 16%—and matches state of the art on five generation evaluation metrics.

## 1 Introduction

Accurate imputation of missing values is essential in real-world tabular data, such as electronic health records (EHR), e-commerce logs, and mobility-sensor streams. High-fidelity synthetic data tables are required in privacy regulations and data-sharing agreements, so that organizations can share information without exposing raw data (Donders et al., 2006; Lin & Tsai, 2020; Assefa et al., 2020; Hernandez et al., 2022). Recent advances show that diffusion models are effective for both imputation and tabular synthesis. Works such as TabSyn, TabDiff, and TabNAT show that a generator trained for synthesis can also reconstruct missing entries, suggesting the feasibility of a single model for both tasks (Zhang et al., 2024; Shi et al., 2025; Zhang et al., 2025b; Lugmayr et al., 2022).

Despite this promise, the objectives and evaluations differ: imputation is judged by pointwise accuracy, whereas synthesis emphasizes diversity and fidelity (Jarrett et al., 2022; Alaa et al., 2022; Zhang et al., 2024). Moreover, stochastic diffusion sampling explores many reverse-time trajectories; naive samples therefore exhibit variance that harms pointwise estimates (Liu et al., 2024; Chen et al., 2024). Existing remedies follow two lines. (i) Multiple-imputation averaging improves accuracy by averaging many samples, but incurs high latency (Zheng & Charoenphakdee, 2022; Zhang et al., 2025a). (ii) Trajectory focusing shapes rules or losses to keep the sampler on a low-noise path, yielding fast and accurate estimates at the cost of reduced distributional coverage (Liu et al., 2024; Chen et al., 2024). Even with strong backbones and mixed-type diffusion, residual mismatch between generated and empirical distributions at sampling time remains a challenge, motivating explicit sampling control.

We propose ImpuGen, a conditional diffusion model that unifies imputation and tabular synthesis via task-aligned sampling. For a practical unified model, imputation must deliver accurate pointwise estimates with low latency, while synthesis must retain coverage of the empirical distribution. ImpuGen addresses both requirements with two sampling strategies:

- Zero-start sampling. The reverse process starts from $x_T = 0$ and conditions on observed entries, producing deterministic imputations without sample averaging.

- Distribution-matching refinement (DMR). After drawing an initial sample, each column is randomly remasked with probability $p$; the model regenerates the masked entries.

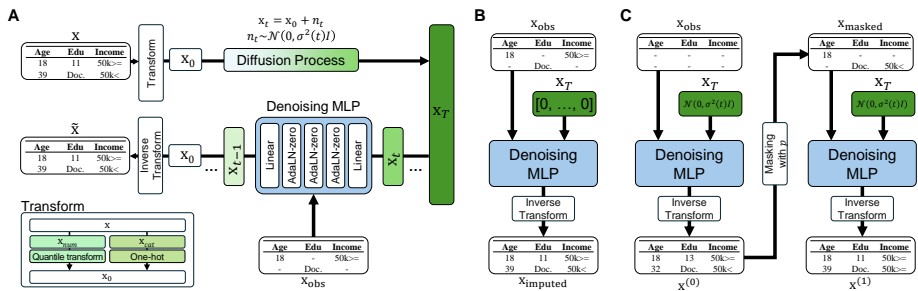

Figure 1: Overview of IMPUGEN. (A) Conditional diffusion backbone. Tabular inputs are quantile-transformed or one-hot-encoded, concatenated into $\mathbf{x}_0$, diffused in EDM $\sigma$-space to $\mathbf{x}_T$, then denoised by an AdaLN-Zero MLP conditioned on $\mathbf{x}_{\text{obs}}$. (B) Zero-start sampling. The reverse process starts from the zero vector $\mathbf{x}_T = \mathbf{0}$. A single reverse pass yields deterministic imputations $\mathbf{x}_{\text{imputed}}$. (C) Distribution-Matching Refinement. From an initial sample $\mathbf{x}^{(0)}$, each column is randomly masked with probability $p$. The masked entries are regenerated once conditioned on the unmasked context, producing $\mathbf{x}^{(1)}$ that better matches the empirical data distribution while maintaining cross-column structure.

Under standard protocols on nine public datasets, IMPUGEN outperforms eleven imputation baselines in pointwise accuracy and matches leading methods on diversity- and fidelity-oriented synthesis metrics. Ablations attribute the gains primarily to the two sampling strategies.

## 2 RELATED WORK

Research on diffusion for tabular data has advanced along two branches: imputation, which fills missing values, and synthesis, which produces realistic tables under privacy constraints.

**Imputation**. TabCSDI (Zheng & Charoenphakdee, 2022) separates observed and missing parts, trains a conditional diffusion model, and performs multiple imputations. SimpDM (Liu et al., 2024) introduces self-supervised alignment and attains high pointwise accuracy with a single reverse process. KnewImp (Chen et al., 2024) reframes diffusion as a Wasserstein gradient flow and modulates diversity with a negative-entropy regularizer. Diffputer (Zhang et al., 2025a) couples diffusion with an EM loop, averaging repainting samples during E-steps and re-estimating the joint in M-steps. MissDiff (Ouyang et al., 2023) trains directly on partially observed tables, modeling the joint without a preliminary imputation stage.

**Synthesis**. TabDDPM (Kotelnikov et al., 2023) and CoDi (Lee et al., 2023) decouple continuous and categorical variables and run two diffusion processes to generate mixed-type tables. TabSyn (Zhang et al., 2024) adopts a two-stage VAE–diffusion pipeline and shows that latent embeddings work well for heterogeneous data. TabDiff (Shi et al., 2025) learns a trainable diffusion schedule with a corrective sampler to mitigate decoding errors. TabNAT (Zhang et al., 2025b) integrates a bidirectional masked transformer with diffusion to obtain an autoregressive-style backbone. Although trained on fully observed data, TabSyn, TabDiff, and TabNAT can also perform imputation via repainting (Lugmayr et al., 2022).

**Remaining gaps**. Methods that reduce error by averaging samples (e.g., TabCSDI, Diffputer) incur latency due to repeated reverse steps and repainting, whereas trajectory-focusing methods (e.g., SimpDM, KnewImp) improve speed and variance at the risk of reduced coverage. Representation and backbone improvements (e.g., TabSyn, TabDDPM, TabDiff, TabNAT) raise overall quality, yet they do not adjust sampling to reduce mismatch to the empirical data distribution.

We address these challenges with IMPUGEN. Zero-start sampling provides deterministic imputations in a single reverse process, avoiding repeated repainting while maintaining accuracy and low latency. DMR sampling performs one round of probability-$p$ re-masking and conditional regeneration, moving samples toward the empirical distribution while preserving cross-column dependencies across continuous and categorical variables.

## 3   METHOD

Figure 1 overviews IMPUGEN. We keep the backbone and training fixed and adjust only the reverse process so that (i) comparisons are fair under a small, fixed number of function evaluations (NFE) and (ii) latency is predictable. Concretely, every sampler uses the same deterministic reverse pass with $T = 50$ steps unless stated otherwise. We align inference to two tasks: imputation targets low-variance point accuracy, while synthesis targets distributional agreement with preserved cross-column structure.

### 3.1   PROBLEM SETUP

Let $\mathcal{D} = \{\mathbf{x}^{(n)}\}_{n=1}^{N}$ be a $D$-column tabular dataset that may contain missing values. Each row $\mathbf{x} \in \mathbb{R}^D$ has a binary mask $\mathbf{m} \in \{0,1\}^D$, where $m_i = 1$ indicates an observed entry. Denote $\mathbf{x}_{\text{obs}} = \mathbf{x} \odot \mathbf{m}$ and $\mathbf{x}_{\text{miss}} = \mathbf{x} \odot (1 - \mathbf{m})$.

**Imputation.**   Given $(\mathbf{x}_{\text{obs}}, \mathbf{m}) \sim P_{\text{data}}$, impute plausible $\mathbf{x}_{\text{miss}}$; the primary metrics are MAE/RMSE.

**Tabular synthesis.**   Sample $\tilde{\mathbf{x}}$ whose distribution matches $P_{\text{data}}$ in fidelity, coverage, downstream utility, and privacy-oriented criteria.

**Unified conditional view.**   Both tasks use the same conditional model

$$p_\theta(\mathbf{x} \mid \mathbf{x}_{\text{obs}}, \mathbf{m}),$$

where imputation draws $\mathbf{x}_{\text{miss}} \sim p_\theta(\mathbf{x}_{\text{miss}} \mid \mathbf{x}_{\text{obs}}, \mathbf{m})$ and unconditional generation sets $\mathbf{m} = \mathbf{0}$ to sample $\mathbf{x} \sim p_\theta(\mathbf{x})$.

### 3.2   DETERMINISTIC EDM BACKBONE

Following the deterministic variant of EDM, we work in $\sigma$-space with a monotone decreasing schedule $\sigma : [0,1] \to [\sigma_{\max}, \sigma_{\min}]$. For a clean row $\mathbf{x}_0$,

$$\mathbf{x}_\tau = \mathbf{x}_0 + \sigma(\tau)\,\varepsilon, \quad \varepsilon \sim \mathcal{N}(0, I).$$

At inference, we integrate the reverse ODE

$$\frac{d\mathbf{x}}{d\tau} = -\,\sigma(\tau)\, s_\theta\big(\mathbf{x}, \sigma(\tau) \mid \mathbf{x}_{\text{obs}}, \mathbf{m}\big),$$

where the conditional score $s_\theta \approx \nabla_{\mathbf{x}} \log p_\sigma(\mathbf{x} \mid \mathbf{x}_{\text{obs}}, \mathbf{m})$ is shared by imputation and synthesis. We use Heun's second-order solver with $T = 50$ or 25 steps.

**Network and encodings.**   Columns are quantile-transformed (continuous) or one-hot-encoded (categorical) and concatenated to form $\mathbf{x}$. Conditioning on $(\mathbf{x}_{\text{obs}}, \mathbf{m})$ enters the AdaLN-Zero MLP via feature-wise affine modulation.

**Training objective.**   We train $s_\theta$ with masked denoising score matching (as in MissDiff):

$$\mathcal{L}_{\text{DSM}} = \big\| \big( s_\theta(\mathbf{x}_\tau, \sigma(\tau) \mid \mathbf{x}_{\text{obs}}, \mathbf{m}) - \varepsilon \big) \odot \mathbf{m} \big\|_2^2, \quad \varepsilon \sim \mathcal{N}(0, I).$$

During training we use the dataset-provided mask $\mathbf{m}$ for each row; for fully observed rows we set $\mathbf{m} = \mathbf{1}^D$. No additional mask mixing is applied.

### 3.3   TASK-ALIGNED INFERENCE OBJECTIVES

We phrase inference-time goals directly in terms of evaluation metrics under a fixed compute budget:

$$\min_{\text{sampler}} \mathbb{E}[|\hat{x} - x|] \quad \text{(imputation)}, \qquad \min_{\text{sampler}} \Delta(q_{\text{gen}}, P_{\text{data}}) \quad \text{(synthesis)},$$

where $\hat{x}$ is a single-pass estimate and $\Delta$ is a distributional discrepancy (e.g., sum of per-column KS distances, energy/C2ST-style scores when available).

## 3.4 ZERO-START SAMPLING FOR IMPUTATION

**Rationale.** Optimizing point-wise error without resorting to multiple-imputation averaging requires a good initialization for the reverse ODE. We therefore initialize at the origin, $\mathbf{x}_T = \mathbf{0}$, which removes stochasticity from the terminal state and yields a single, deterministic reverse pass aligned with the $L_1/L_2$ error objective.

**Empirical effect.** Compared with multiple-imputation averaging under the same NFE, zero-start produces lower MAE and RMSE across benchmarks while preserving latency predictability (see Fig. 4). In practice, this makes zero-start a favorable default when the evaluation metric is a single-pass point estimate.

**Theoretical support.** In the one-dimensional conditional setting, we prove that the zero-start reverse flow converges to the conditional median (Appendix, Thm. A). Since the median minimizes $L_1$ risk, this explains the observed reduction in absolute error without requiring ensembling over multiple terminal draws.

**Procedure.** We run one deterministic reverse pass with $(\mathbf{x}_{\text{obs}}, \mathbf{m})$ held fixed and $\mathbf{x}_T = \mathbf{0}$, and take the resulting $\hat{\mathbf{x}}$ as point estimates for $\mathbf{x}_{\text{miss}}$.

---

**Algorithm 1** Zero-Start Imputation

---

1: **Inputs:** reverse schedule $\{\sigma_t\}_{t=T}^0$, observed pair $(\mathbf{x}_{\text{obs}}, \mathbf{m})$
2: Initialize $\mathbf{x}_T \leftarrow \mathbf{0}$
3: $\hat{\mathbf{x}} \leftarrow \text{EDM}(\mathbf{x}_T; \mathbf{x}_{\text{obs}}, \mathbf{m})$
4: **return** $\hat{\mathbf{x}} \odot (1 - \mathbf{m})$ as imputed entries

---

## 3.5 DISTRIBUTION-MATCHING REFINEMENT (DMR) FOR SYNTHESIS

**Motivation.** Even with mixed-type diffusion backbones, samples from a single reverse pass can exhibit marginal or conditional drift relative to $P_{\text{data}}$. DMR adds a one-shot, training-free refinement that nudges the sample distribution toward the empirical one while keeping cross-column dependencies learned by the backbone.

**Single-step DMR.** From an initial synthetic row $\mathbf{x}^{(0)}$ generated with a standard pass (draw $\mathbf{x}_T \sim \mathcal{N}(0, \sigma_{\max}^2 I)$, then reverse ODE), independently mask each column with probability $p$ to form $\mathbf{m}^{\text{dmr}}$. Run one conditional reverse pass that regenerates only the masked entries given the unmasked context:

$$\mathbf{x}^{\text{ref}} \leftarrow \text{EDM}\Big(\mathbf{x}^{(0)}; \mathbf{x}_{\text{obs}} = \mathbf{x}^{(0)} \odot (1 - \mathbf{m}^{\text{dmr}}), \mathbf{m} = \mathbf{m}^{\text{dmr}}\Big).$$

To match the baseline compute budget, we set $T=25$ steps for the initial pass and $T=25$ for the refinement pass (total NFE equal to a single $T=50$ pass). This single refinement reduces energy distance with minimal compute.

---

**Algorithm 2** Distribution-Matching Refinement (DMR)

---

1: **Inputs:** reverse schedule $\{\sigma_t\}_{t=T}^0$, mask probability $p$
2: $\mathbf{x}_T \sim \mathcal{N}(0, \sigma_{\max}^2 I)$, $\quad \mathbf{x}^{(0)} \leftarrow \text{EDM}(\mathbf{x}_T; \mathbf{m} = \mathbf{0})$
3: Sample $\mathbf{m}^{\text{dmr}} \sim \text{Bernoulli}(p)^{\otimes D}$
4: $\mathbf{x}^{\text{ref}} \leftarrow \text{EDM}\Big(\mathbf{x}^{(0)}; \mathbf{x}_{\text{obs}} = \mathbf{x}^{(0)} \odot (1 - \mathbf{m}^{\text{dmr}}), \mathbf{m} = \mathbf{m}^{\text{dmr}}\Big)$
5: **return** $\mathbf{x}^{\text{ref}}$

---

**Choice of $p$.** After training, we choose $p$ on a validation split via a simple grid search:

$$p^\star = \arg\min_{p \in \{0.1, 0.2, \ldots, 0.9\}} \text{ED}\big(q_{\text{gen}}(p), P_{\text{data}}\big),$$

| Dataset | # Train | # Test | # Num | # Cat | Task |
|---|---|---|---|---|---|
| Bean | 12,249 | 1,362 | 16 | 1 | Classification |
| Gesture | 8,569 | 953 | 33 | 1 | Classification |
| Housing | 18,576 | 2,064 | 9 | 0 | Regression |
| Letter | 18,000 | 2,000 | 16 | 1 | Classification |
| Magic | 17,117 | 1,902 | 10 | 1 | Classification |
| Adult | 32,651 | 16,281 | 9 | 6 | Classification |
| Default | 27,000 | 3,000 | 14 | 10 | Classification |
| News | 35,679 | 3,965 | 46 | 2 | Regression |
| Shoppers | 11,097 | 1,233 | 10 | 8 | Classification |

Table 1: Descriptions of the nine benchmark datasets. # Num and # Cat denote the number of numerical and categorical columns.

where ED denotes the energy distance between generated and empirical distributions. We keep the selected $p^\star$ fixed for all subsequent evaluations. As a baseline, let $ED_{base}$ be the energy distance from a single reverse pass with $T{=}50$ (no DMR). If the best grid value does not reduce the energy distance relative to $ED_{base}$, we omit DMR and use the single $T{=}50$ pass.

# 4 EXPERIMENTS

## 4.1 EXPERIMENTAL SETTINGS

This section outlines the experimental protocol used to evaluate IMPUGEN. We first describe the datasets, baselines, and evaluation metrics, then present quantitative results for imputation and synthesis, followed by ablation studies.

**Datasets.** Following prior work (Zhang et al., 2025a), we use nine publicly available datasets—Adult, Bean, Default, Gesture, Housing, Letter, Magic, News, and Shoppers (Asuncion et al., 2007; Pace & Barry, 1997). Detailed information for each dataset, including the number of rows and columns are summarized in Table 1.

**Baseline imputation models.** We compare IMPUGEN with eleven baselines, organized into five categories: Diffusion: DiffPuter (Zhang et al., 2025a), SimpDM (Liu et al., 2024), KnewImp (Chen et al., 2024); Transformer: ReMasker (Du et al., 2024), MaCoDE (An et al., 2025); Iterative: HyperImpute (Jarrett et al., 2022), MissForest (Stekhoven & Bühlmann, 2012), EM (Dempster et al., 1977), MICE (Van Buuren & Groothuis-Oudshoorn, 2011); GAN: GAIN (Yoon et al., 2018). Graph: GRAPE (You et al., 2020).

**Baseline generation models.** For synthetic-table generation we benchmark against seven generators drawn from four categories: Diffusion: TabDDPM (Kotelnikov et al., 2023), TabSyn (Zhang et al., 2024), TabDiff (Shi et al., 2025), TabNAT (Zhang et al., 2025b); Transformer: MaCoDE; GAN: CTGAN (Xu et al., 2019); VAE: TVAE (Xu et al., 2019).

**Data splits.** We used an 90:10 train and test split ratio to evaluate our model, except for Adult, which follows its official UCI split. If a validation set is required, we hold out 10% of the training set.

For dDistance-to-Closest-Record (DCR), each dataset is divided 50% for training and 50% for testing. In the case of Data-Plagiarism Index Membership-Inference Attack (DPI–MIA)(Ward et al., 2024), the data are further partitioned into 50% training, 25% hold-out, and 25% reference subsets.

**Missingness generation.** We reproduce the three masking mechanisms used in prior work(Zhao et al., 2023; Zhang et al., 2025a): (i) missing completely at random (MCAR): each cell is masked independently with probability $r$; (ii) missing at random (MAR): a subset of fully observed features is sampled, and a logistic model is fitted to generate masks for the remaining columns; (iii) missing

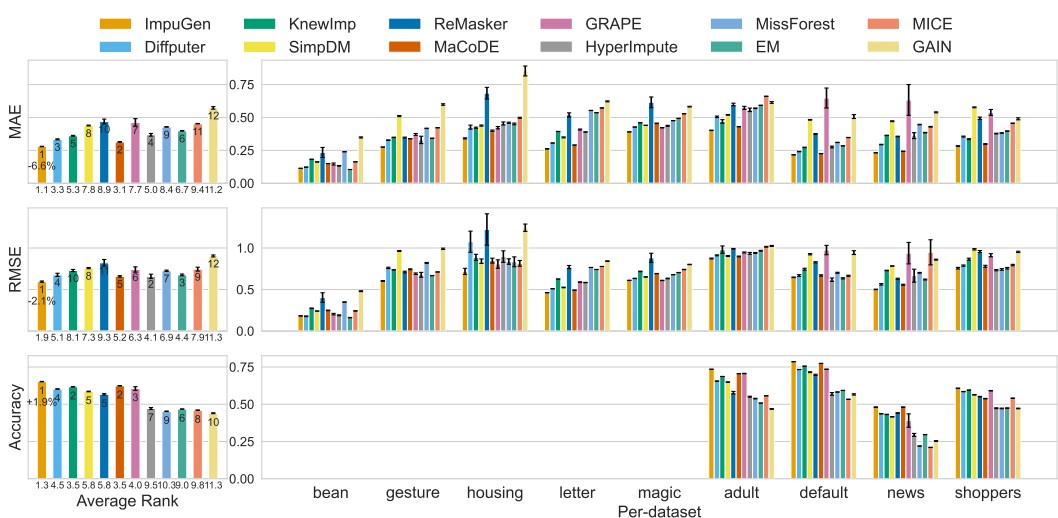

Figure 2: **MCAR 30% imputation performance on nine datasets.** We compare IMPUGEN with eleven baselines on continuous (MAE, RMSE) and categorical (Accuracy) columns. **Left mini-panel:** for each metric, the bar height is the mean score across the nine datasets; the x-tick label underneath the bar is the corresponding average rank, and the numeral printed inside the bar is that rank value itself. The percentage below first bar indicates the averaged relative change of IMPUGEN over the best competing baseline. On average, IMPUGEN reduces MAE by **6.6%** and RMSE by **2.1%**, while also achieving the top categorical-accuracy rank.

not at random (MNAR): features are split into two groups, the first group feeds a logistic model that determines the mask of the second, and MCAR is then applied to the first.

We test missing rates $r \in \{30, 50, 70\%\}$; the main paper reports results for MCAR at 30%.

**Imputation metrics.** For continuous features, we compute column-wise MAE and RMSE, then average the scores across all columns. For categorical features, we measure classification accuracy. Stochastic baselines (MICE, MaCoDE, and DiffPuter) are evaluated with multiple imputation: ten stochastic draws are generated for each missing entry, averaged, and then scored.

**Generation metrics.** We follow the TabSyn protocol (Zhang et al., 2024) and assess the three aspects of synthetic–data quality—Fidelity, Utility, and Privacy.

**Fidelity**. We report five distributional scores: (i) Shape—the Kolmogorov–Smirnov statistic between the marginal density of each column and its synthetic counterpart; (ii) Trend—the deviation in pair-wise correlations (Pearson for continuous columns, total-variation distance for categorical ones); (iii) $\alpha$-precision—the fraction of synthetic samples whose nearest real neighbor lies within the $\alpha$-quantile radius of the real manifold; (iv) $\beta$-recall—the coverage of the real manifold by synthetic samples; and (v) C2ST—the accuracy of a logistic discriminator trained to distinguish real from synthetic rows. All scores range from 0 to 1; higher values indicate better performance.

**Utility**. Following TabSyn (Zhang et al., 2024), we train one predictor on the synthetic set and another on the real training set, and evaluate both on the real test set. For evaluating classification performance, we used macro-AUROC ratio, defined as the score of the synthetic-trained model divided by that of the real-trained model. For comparing regression performance, we use the inverse RMSE ratio, the real-trained RMSE divided by the synthetic-trained RMSE. A ratio greater than one means that learning from synthetic data matches or exceeds the performance obtained with real data.

**Privacy metrics**. We evaluate privacy with two complementary scores.

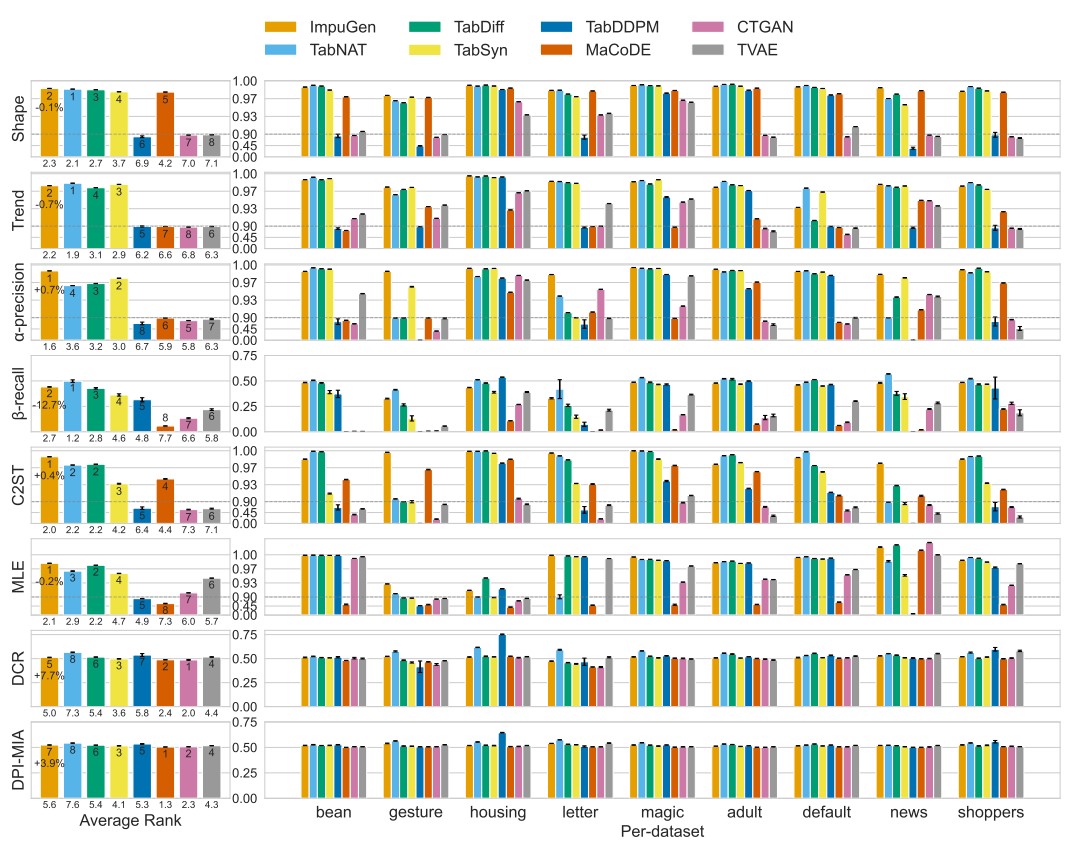

Figure 3: **Synthetic-table generation quality.** We benchmark IMPUGEN against seven baselines on fidelity ($\alpha$-precision), downstream utility (MLE), and privacy (C2ST, DPI–MIA). **Left mini-panel:** bar height is the mean score across datasets; the x-tick shows the corresponding average rank, and the numeral inside each bar repeats that rank. Percentages beneath the bars report IMPUGEN's macro-average change relative to the best baseline. **Right panel:** per-dataset bars supporting those averages. To visualise saturated metrics more clearly, the $y$-axis uses a broken scale: values below 0.90 are compressed, whereas 0.90–1.00 are expanded. Overall, IMPUGEN matches or exceeds the latest SOTA, improving $\alpha$-precision by **0.7 %**.

DCR. For every synthetic sample, we verify whether its nearest real neighbor comes from the training split or the hold-out test split; under an even 50:50 split, an ideal generator achieves a DCR of 0.5.

DPI-MIA. Following Ward et al. (2024) (Ward et al., 2024), we split the real data into train, hold-out, and reference sets. For each real point we compute a data-plagiarism index (DPI) based on its $k$-nearest neighbours, where $k$ ranges from 1 to 30. A membership attacker is then evaluated for every $k$; we report the largest AUROC achieved across this sweep. Lower AUROC values correspond to stronger privacy.

See Appendix B for detailed information.

**Label handling.** Multiple-purpose models such as IMPUGEN and MaCoDE use the label column during training to learn the full joint distribution but mask it at test time, thereby preventing label leakage during evaluation. Pure imputation baselines never observe labels. This was examined more closely in Figure 6.

**Repetition and seeds.** All reported numbers are averaged over five independent runs. We fix the random seed to $\{0, 1, 2, 3, 4\}$ in turn for missing mask generation, weight initialization, and any stochastic components.

| Model | Params (M) | Train (s) | Imp. (s) | Gen. (s) |
|---|---|---|---|---|
| ImpuGen | 10.2 | 408 | 2.9 | 2.9 |
| MaCoDE | 1.3 | 320 | 6.2 | 0.7 |
| Diffputer | 10.6 | 2,937 | 307.1 | – |
| SimpDM | 4.5 | 408 | 1.0 | – |
| KnewImp | 0.09 | 47 | 7.2 | – |
| ReMasker | 0.7 | 4,858 | 0.8 | – |
| TabSyn$^\dagger$ | 10.6 | 1,382 | – | 1.6 |
| TabDiff | 10.6 | 3,112 | – | 14.5 |
| TabNAT | 13.3 | 4,981 | – | 20.3 |
| CTGAN | 21.7 | 7,459 | – | 7.6 |
| TVAE | 9.6 | 499 | – | 0.5 |

Table 2: Parameter count and wall-clock time on the `Adult` dataset with RTX 5090 GPU. **Imp.**: imputation; **Gen.**: generation; "–": not applicable.
$^\dagger$TabSyn = VAE (1,009s) + diffusion (373s).

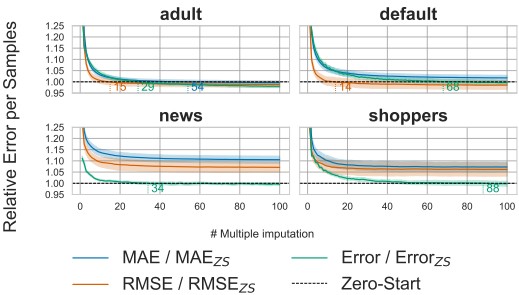

Figure 4: **Zero-start vs. multiple imputation.** For each dataset, the curve shows the ratio of MI with $k$ samples to zero-start performance ($k = 1$–$100$)

## 4.2 MISSING-VALUE IMPUTATION

**Main Results.** Figure 2 shows that under MCAR 30% IMPUGEN ranks first on MAE, RMSE, and categorical accuracy, reducing MAE by 6.6% compared to the strongest baseline. Similar results are shown with MAR and MNAR with 30, 50, 70%. The consistency across nine masking regimes indicates that the zero-start sampling generalizes to severe sparsity.

**Runtime and model size.** Table 2 reports wall-clock runtime and parameter count on the `Adult` dataset. Although IMPUGEN has the same parameter number as DiffPuter, it trains in only 408s and completes imputation in 2.9s. The gain comes from the deterministic zero-start strategy, which removes multi-sample repainting procedure, which can be time-consuming.

## 4.3 TABULAR DATA SYNTHESIS

**Fidelity.** Figure 3 reports five fidelity metrics: Shape, Trend, $\alpha$-precision, $\beta$-recall, and C2ST. On Shape and Trend, IMPUGEN achieves better ranks on continuous-dominant tables (`Gesture`, `Housing`), whereas TabNAT scores better on categorical-dominant datasets (`Adult`, `Shoppers`). However, the average ranks differ only slightly—2.3 vs. 2.1 for Shape and 2.2 vs. 1.9 for Trend. For $\alpha$-precision, IMPUGEN records the best mean rank of 1.6, consistent with the gains from DMR sampling. TabNAT tops $\beta$-recall, possibly due to its larger parameter budget (13.3 M parameters, the largest among the diffusion models). On C2ST, IMPUGEN attains a mean rank of 2.0, narrowly ahead of TabNAT and TabDiff. Overall, IMPUGEN shows a clear advantage in $\alpha$-precision and remains competitive on the other fidelity metrics.

**Down-stream utility.** On the MLE benchmark, IMPUGEN attains an average rank of 2.1, slightly ahead of TabDiff at 2.2.

**Privacy.**    In terms of DCR, most diffusion models cluster around the ideal value of 0.5, whereas TabNAT scores noticeably higher than the rest. IMPUGEN records an average rank of 5.0, second only to TabSyn within the diffusion group. In DPI–MIA, TabSyn leads the diffusion models with a rank of 4.1, followed by TabDiff at 5.4 and IMPUGEN at 5.6, while TabNAT trails at 7.6. Privacy scores tend to move in the opposite direction of $\beta$-recall: models such as MaCoDE and CTGAN, which obtain low $\beta$-recall, show relatively strong privacy. This pattern suggests that aggressive coverage of sparse regions in the data distribution may increase privacy-leakage risk.

**Runtime and Model Size.**    IMPUGEN generates the `Adult` dataset in 2.9 seconds—much faster than TabNAT's 20.3 seconds and between TabSyn (1.6s) and TabDiff (14.5s).

### 4.4   ABLATION STUDIES

**Zero-start vs. multiple imputation.**    We compared zero-start sampling with multiple imputation (MI) by generating up to 100 MI samples for each missing row. On `Adult`, MI needed at least 54 samples to match the MAE of zero-start. On `Default`, `News`, and `Shoppers`, the MAE gap was still present after 100 samples. MAE shows the largest difference, which is consistent with one-dimensional result that zero-start converges to the posterior median during the reverse process. Taken together, these results show that IMPUGEN goes beyond a simple combination of TabCSDI and MissDiff. By adopting a more efficient sampling strategy, it improves both imputation accuracy and runtime. (Zheng & Charoenphakdee, 2022; Ouyang et al., 2023)

**DMR sampling.**    In Figure 5, on `Bean`, `Gesture`, `Letter`, `Adult`, `Default`, `News`, and `Shoppers`, we compare IMPUGEN with and without distribution-matching refinement (DMR) sampling. Adding DMR increases $\alpha$-precision by 1.7% and yields consistent improvements across the fidelity suite (Trend +0.3%, C2ST +0.6%, MLE +0.4%). Overall, DMR sampling enhances not only $\alpha$-precision but also distributional fidelity.

**Label leakage impact.**    To assess the effect of label leakage, we retrained IMPUGEN after removing the label columns. Including labels increases MAE and RMSE by 0.4% and 2.3%, respectively, indicating that label information slightly degrades imputation accuracy instead of improving it.

## 5   CONCLUSION

In this paper, We introduced IMPUGEN, a unified framework for imputation and tabular data synthesis that employs two task-aligned sampling strategies. zero-start sampling removes the trade-off between speed and diversity that affects diffusion-based imputers and yields accurate pointwise accuracy in a single reverse process. DMR sampling is column-agnostic and further enhances overall fidelity.

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

**Structure at a Glance**   The appendix is divided into four self-contained parts: **Section A** formalizes all evaluation metrics used in the main paper; **Section B** presents visualizations from the ablation studies, focusing on distribution-matching refinement (DMR) sampling and label leakage control; **Section C** provides complete implementation details for all baselines; and **Section D** expands the imputation benchmarks to both in-sample and out-of-sample settings under MCAR, MAR, and MNAR masks at 30 %, 50 %, and 70 % missingness.

# A   THEORETICAL ANALYSIS

## A.1   ZERO-START SAMPLING

**Assumption A1 (Finite Lipschitz constant).**   Let $p_{\text{data}}$ be the data distribution. For a score $s_\theta(\cdot, \sigma)$ trained with Gaussian perturbation of standard deviation $\sigma$, its Jacobian Frobenius norm is bounded by

$$\left\| \nabla_x s_\theta(x, \sigma) \right\|_F \leq \frac{C}{\sigma}, \qquad x \sim p_{\text{data}},$$

for some constant $C$ depending on the smoothness of $\log p_\sigma$ (Yang et al., 2023). Since $\sigma(\tau) \geq \sigma_{\min} > 0$ for all $\tau \in [0, 1]$, the reverse field is globally $L_\theta = C/\sigma_{\min}$-Lipschitz.

**One-dimensional posterior-median convergence (sketch).**   Let $\varphi_{T \to 0}(x_T)$ be the deterministic EDM flow. Because the vector field is globally Lipschitz, the flow is unique and strictly monotone in its initial state. With a symmetric terminal distribution $x_T \sim \mathcal{N}(0, \sigma_{\max}^2)$, the monotone map $\varphi_{T \to 0}$ preserves median ordering, so $\varphi_{T \to 0}(0)$ is the median of the conditional distribution of $x_0$ given $x_T$.

**Algorithm.**   At inference, run the deterministic EDM sampler initialized at $\mathbf{x}_T = \mathbf{0}$ and conditioned on $(\mathbf{x}_{\text{obs}}, \mathbf{m})$ through $s_\theta$. A single $T$-step reverse pass yields deterministic imputations that match or surpass multiple-imputation accuracy at a fraction of the latency.

# B   EVALUATION METRICS

## B.1   IMPUTATION METRICS

**Robust macro-averaged error metrics**   Some datasets contain preprocessing errors that yield extreme values. For instance, one row in the `News` dataset was left unnormalized during min–max scaling, producing an exorbitant $z$-score that distorts column-wise error statistics. To mitigate such artifacts without affecting regular observations, we discard any entry whose absolute $z$-score exceeds 100.

For each continuous column $d \in \mathcal{C}$, define the evaluation index set

$$\mathcal{M}_d = \left\{ n \mid m_d^{(n)} = 0, \ |z_d^{(n)}| < 100 \right\},$$

where $m_d^{(n)} = 0$ denotes a missing value at position $(n, d)$. We compute the MAE and RMSE on $\mathcal{M}_d$ for every column individually and then take their simple average over all continuous columns to obtain the *macro-averaged* error, thereby assigning equal weight to each column regardless of scale.

$$\text{MAE}_{\text{macro}} = \frac{1}{|\mathcal{C}|} \sum_{d \in \mathcal{C}} \frac{1}{|\mathcal{M}_d|} \sum_{n \in \mathcal{M}_d} |\hat{z}_d^{(n)} - z_d^{(n)}|, \tag{1}$$

$$\text{RMSE}_{\text{macro}} = \frac{1}{|\mathcal{C}|} \sum_{d \in \mathcal{C}} \sqrt{\frac{1}{|\mathcal{M}_d|} \sum_{n \in \mathcal{M}_d} \left(\hat{z}_d^{(n)} - z_d^{(n)}\right)^2}. \tag{2}$$

## B.2 FIDELITY METRICS

We assess the fidelity of a synthetic table $\mathcal{D}^{\mathrm{s}}$ relative to a real table $\mathcal{D}^{\mathrm{r}}$ using five complementary metrics. All scores are normalized to the range $[0, 1]$; higher values indicate better fidelity. Assume each table contains $D$ columns and $N_{\mathrm{r}}$ and $N_{\mathrm{s}}$ rows, respectively.

**Shape similarity (Wüst, 2011)**    For each continuous column $d \in \mathcal{C}$, we measure the discrepancy with the Kolmogorov–Smirnov (KS) statistic—the supremum of the absolute difference between the empirical cumulative distribution functions of the real and synthetic tables (Massey Jr, 1951):

$$\varepsilon_d^{\mathrm{KS}} = \sup_x \big| \hat{F}_d^{\mathrm{r}}(x) - \hat{F}_d^{\mathrm{s}}(x) \big|. \tag{1}$$

For each categorical column $d \in \mathcal{Q}$, the discrepancy is the total-variation distance (TVD) between the corresponding empirical probability mass functions:

$$\varepsilon_d^{\mathrm{TVD}} = \tfrac{1}{2} \sum_{k \in \mathcal{K}_d} \big| p_d^{\mathrm{r}}(k) - p_d^{\mathrm{s}}(k) \big|, \tag{2}$$

where $\mathcal{K}_d$ denotes the set of categories in column $d$.

Finally, we average the per-column discrepancies across all $D$ columns and convert them into a similarity score:

$$S_{\mathrm{shape}} = 1 - \frac{1}{D} \sum_{d=1}^{D} \varepsilon_d. \tag{3}$$

**Trend similarity (Wüst, 2011)**    For every unordered column pair $(d_1, d_2)$ with $d_1 < d_2$ we compute a type-specific discrepancy $\Delta_{d_1 d_2}$.

**Continuous–continuous.** The discrepancy equals half the absolute difference between the sample Pearson correlations of the real and synthetic tables:

$$\Delta_{d_1 d_2}^{\mathrm{Pearson}} = \tfrac{1}{2} \big| \rho_{d_1 d_2}^{\mathrm{r}} - \rho_{d_1 d_2}^{\mathrm{s}} \big|, \tag{4}$$

where $\rho_{d_1 d_2}$ is the Pearson coefficient.

**All other type combinations.** First build an empirical contingency table $C_{d_1 d_2}$. If either column is continuous, discretize it into $K = 20$ equal-frequency bins. The discrepancy is the total-variation distance (TVD) between the real and synthetic contingency tables:

$$\Delta_{d_1 d_2}^{\mathrm{TVD}} = \tfrac{1}{2} \sum_{a \in \mathcal{K}_{d_1}} \sum_{b \in \mathcal{K}_{d_2}} \big| C_{d_1 d_2}^{\mathrm{r}}(a, b) - C_{d_1 d_2}^{\mathrm{s}}(a, b) \big|, \tag{5}$$

where $\mathcal{K}_d$ denotes the category set (or bins) of column $d$.

Finally, average the discrepancies over all $\binom{D}{2}$ unordered pairs and convert them to a similarity score:

$$S_{\mathrm{trend}} = 1 - \frac{2}{D(D-1)} \sum_{d_1 < d_2} \Delta_{d_1 d_2}. \tag{6}$$

**$\alpha$-precision (Alaa et al., 2022)**    We measure how tightly synthetic rows occupy the high-density region of the real table.

**Embedding and distance.** Each row $\mathbf{d}$ is mapped to $\phi(\mathbf{d}) \in [0, 1]^H$ by min–max scaling the continuous features and one-hot encoding the categorical features. The distance between two rows is Euclidean:

$$E(\mathbf{a}, \mathbf{b}) = \| \phi(\mathbf{a}) - \phi(\mathbf{b}) \|_2. \tag{7}$$

**Real-data center and radii.**

$$\mathbf{c} = \frac{1}{N_{\mathrm{r}}} \sum_{n=1}^{N_{\mathrm{r}}} \phi\big( \mathbf{d}_n^{\mathrm{r}} \big), \tag{8}$$

$$R(\alpha_k) = \mathrm{quantile}_{\alpha_k} \Big\{ E\big( \mathbf{d}_n^{\mathrm{r}}, \mathbf{c} \big) \Big\}_{n=1}^{N_{\mathrm{r}}}. \tag{9}$$

The grid $\alpha_k = (k-1)/29, \; k = 1, \ldots, 30$ matches the reference implementation.

**Precision curve.** For each $\alpha_k$, the share of synthetic rows within the corresponding radius is

$$p(\alpha_k) = \frac{1}{N_\mathrm{s}} \sum_{n=1}^{N_\mathrm{s}} \mathbf{1}\big[E(\mathbf{d}_n^\mathrm{s}, \mathbf{c}) \leq R(\alpha_k)\big]. \tag{10}$$

**Summary statistic.** The deviation from the ideal diagonal $p(\alpha) = \alpha$ is converted to a similarity score:

$$S_\mathrm{precision} = 1 - \frac{\sum_{k=1}^{30}\big|\alpha_k - p(\alpha_k)\big|}{\sum_{k=1}^{30} \alpha_k}. \tag{11}$$

**$\beta$-recall (Alaa et al., 2022)** This metric complements $\alpha$-precision by checking whether every real row is represented by a sufficiently close synthetic neighbor.

**(i) Real–synthetic match**

$$s^*(\mathbf{d}_n^\mathrm{r}) = \arg \min_{\mathbf{d}^\mathrm{s} \in \mathcal{D}^\mathrm{s}} E\big(\mathbf{d}_n^\mathrm{r}, \mathbf{d}^\mathrm{s}\big), \tag{12}$$

$$d_\mathrm{rs}(\mathbf{d}_n^\mathrm{r}) = E\big(\mathbf{d}_n^\mathrm{r}, s^*(\mathbf{d}_n^\mathrm{r})\big). \tag{13}$$

**(ii) Real–real reference**

$$d_\mathrm{rr}(\mathbf{d}_n^\mathrm{r}) = \min_{\substack{m=1 \\ m \neq n}}^{N_\mathrm{r}} E\big(\mathbf{d}_n^\mathrm{r}, \mathbf{d}_m^\mathrm{r}\big). \tag{14}$$

**(iii) Synthetic radii**

$$\mathbf{c}_\mathrm{s} = \frac{1}{N_\mathrm{s}} \sum_{m=1}^{N_\mathrm{s}} \phi\big(\mathbf{d}_m^\mathrm{s}\big), \tag{15}$$

$$R_\mathrm{s}(\alpha_k) = \mathrm{quantile}_{\alpha_k}\Big\{E\big(s^*(\mathbf{d}_n^\mathrm{r}), \mathbf{c}_\mathrm{s}\big)\Big\}_{n=1}^{N_\mathrm{r}}. \tag{16}$$

**(iv) Coverage curve**

$$b(\alpha_k) = \frac{1}{N_\mathrm{r}} \sum_{n=1}^{N_\mathrm{r}} \mathbf{1}\Big[d_\mathrm{rs}\big(\mathbf{d}_n^\mathrm{r}\big) \leq d_\mathrm{rr}\big(\mathbf{d}_n^\mathrm{r}\big) \wedge$$
$$E\big(s^*(\mathbf{d}_n^\mathrm{r}), \mathbf{c}_\mathrm{s}\big) \leq R_\mathrm{s}(\alpha_k)\Big]. \tag{17}$$

**(v) Summary statistic**

$$S_\mathrm{coverage} = 1 - \frac{\sum_{k=1}^{30}\big|\alpha_k - b(\alpha_k)\big|}{\sum_{k=1}^{30} \alpha_k}. \tag{18}$$

Both $S_\mathrm{precision}$ and $S_\mathrm{coverage}$ reach 1 when their curves coincide with the diagonal and decrease as deviations grow, yielding single-number summaries of fidelity ($\alpha$) and coverage ($\beta$).

**Classifier two-sample test (C2ST) (Wüst, 2011)** A logistic regression classifier is trained to distinguish the union of the real and synthetic tables, $\mathcal{D}^\mathrm{r} \cup \mathcal{D}^\mathrm{s}$. Following the SDMetrics implementation, we employ three-fold stratified cross-validation: in each fold $k$ ($k = 1, 2, 3$) the model is fitted on two folds and evaluated on the held-out fold, yielding an AUROC score $\mathrm{AUROC}_k$.

**Detection power**

$$d_k = \max\{0,\ 2\,\mathrm{AUROC}_k - 1\}, \tag{19}$$

which equals $0$ when the classifier performs no better than chance ($\mathrm{AUROC}_k \leq 0.5$) and rises linearly to $1$ under perfect separability.

**Similarity score**

$$S_{\mathrm{C2ST}} = 1 - \frac{1}{3}\sum_{k=1}^{3} d_k. \tag{20}$$

Thus $S_{\mathrm{C2ST}} = 1$ when the discriminator cannot distinguish the two tables at all, and $S_{\mathrm{C2ST}} = 0$ under perfect separation.

## B.3  UTILITY METRIC

**Machine-learning efficiency (MLE)**   This metric quantifies how much predictive utility is retained when a model is trained on the synthetic table instead of the real one.

**Protocol**

1. *Split.* The real table $\mathcal{D}^{\mathrm{r}}$ is divided once into an $8{:}1$ train–validation split (the split is stratified for classification targets).

2. *Model search on real data.* A grid search selects hyperparameters that maximize the macro-AUROC (classification) or minimize the RMSE (regression) on the validation set, yielding a score $s_{\mathrm{real}}$.

3. *Model search on synthetic data.* The same search is repeated on a synthetic table $\mathcal{D}^{\mathrm{s}}$ of equal size, producing $s_{\mathrm{syn}}$.

**Score**

$$S_{\mathrm{MLE}} = \begin{cases} \dfrac{s_{\mathrm{syn}}}{s_{\mathrm{real}}}, & \text{classification}, \\[2mm] \dfrac{s_{\mathrm{real}}}{s_{\mathrm{syn}}}, & \text{regression}. \end{cases} \tag{21}$$

**Notes.**

- TABPFN grids are skipped whenever the number of rows exceeds the model's built-in limit of $10{,}000$ rows.

- Each grid search evaluates exactly one train/validation split; the best configuration is refit on the full split before computing $s_{\mathrm{real}}$ or $s_{\mathrm{syn}}$.

**Hyper-parameter grids.**

| XGBClassifier | |
|---|---:|
| Parameter | Values |
| n_estimators | 10, 50, 100 |
| min_child_weight | 1, 10 |
| max_depth | 5, 10, 20 |
| gamma | 0.0, 1.0 |
| objective | `binary:logistic` |
| tree_method | `hist` |
| device | `cpu` |
| enable_categorical | True |

## B.4  PRIVACY METRICS

**Distance to closest record (DCR) (Zhang et al., 2024)**   For every synthetic row $s \in \mathcal{D}^{\mathrm{s}}$ we compute its Gower distance to the nearest record in the real training split $\mathcal{D}_{\mathrm{train}}$ and in an independent

**TabPFNClassifier**

| Parameter | Values |
|---|---|
| n_estimators | 4, 8, 16 |
| softmax_temperature | 0.8, 0.9, 1.0 |
| balance_probabilities | True |

**XGBRegressor**

| Parameter | Values |
|---|---|
| n_estimators | 10, 50, 100 |
| min_child_weight | 1, 10 |
| max_depth | 5, 10, 20 |
| gamma | 0.0, 1.0 |
| objective | reg:squarederror |
| tree_method | hist |
| device | cpu |
| enable_categorical | True |

**TabPFNRegressor**

| Parameter | Values |
|---|---|
| n_estimators | 4, 8, 16 |
| softmax_temperature | 0.8, 0.9, 1.0 |

hold-out split $\mathcal{D}_{\text{hold}}$:

$$
\begin{aligned}
d_{\text{train}}(s) &= \min_{r \in \mathcal{D}_{\text{train}}} G(s, r), \\
d_{\text{hold}}(s) &= \min_{h \in \mathcal{D}_{\text{hold}}} G(s, h).
\end{aligned}
\tag{22}
$$

The DCR score is the proportion of synthetic rows that lie closer to the training split than to the hold-out split:

$$
\text{DCR} = \frac{1}{|\mathcal{D}^{\text{s}}|} \sum_{s \in \mathcal{D}^{\text{s}}} \mathbf{1}\big[d_{\text{train}}(s) < d_{\text{hold}}(s)\big].
\tag{23}
$$

A value of $\text{DCR} \approx 0.5$ indicates that a synthetic record is equally likely to be nearer to the training split as to the hold-out split, which is a positive privacy signal.

**Data plagiarism index membership-inference attack (DPI-MIA) (Ward et al., 2024)**  The *data plagiarism index* (DPI) for a query record $q$ is the fraction of synthetic rows among its $k$ nearest neighbors (NNs) in a reference pool that mixes real and synthetic data. We sweep $k = 1, \ldots, 30$ and retain the most informative value.

**Reference and query sets**

$$
\mathcal{D}_{\text{ref}} = \mathcal{D}_{\text{train}} \cup \mathcal{D}^{\text{s}}, \qquad \mathcal{D}_{\text{query}} = \mathcal{D}_{\text{hold}} \cup \mathcal{D}_{\text{train}}.
\tag{24}
$$

Rows from $\mathcal{D}^{\text{s}}$ carry label 1 and those from $\mathcal{D}_{\text{train}}$ carry label 0 inside $\mathcal{D}_{\text{ref}}$. In $\mathcal{D}_{\text{query}}$ the ground-truth membership is 0 (hold-out) or 1 (training).

**Per-record DPI at neighborhood size** $k$  Let $g_1, \ldots, g_k \in \{0, 1\}$ be the labels of the $k$ nearest neighbors of $q$ in $\mathcal{D}_{\text{ref}}$ (L1 distance on the min–max + one-hot embedding). The DPI value is

$$
\text{DPI}_k(q) = \frac{1}{k} \sum_{i=1}^{k} g_i.
\tag{25}
$$

**Attack effectiveness**

$$\text{AUC}(k) = \text{ROC-AUC}\Big(\{\text{DPI}_k(q)\}_{q \in \mathcal{D}_{\text{query}}}, \tag{26}$$

$$\{\text{label}(q)\}_{q \in \mathcal{D}_{\text{query}}}\Big), \tag{27}$$

$$\text{DPI-MIA} = \max_{k=1}^{30} \text{AUC}(k). \tag{28}$$

**Interpretation** A value near 0.5 indicates that synthetic rows do not enable the attacker to distinguish training records from unseen hold-out records; higher values provide stronger evidence of data copying.

# C   ABLATION STUDY

This section analyzes two factors: distribution-matching refinement (DMR) sampling for tabular synthesis and the effect of label leakage on imputation.

**Distribution-matchinf refinement (DMR).**   Figure S5 summarizes the impact of DMR On `Bean`, `Gesture`, `Letter`, `Adult`, `Default`, `News`, and `Shoppers`. We therefore compare the full IMPUGEN model with an otherwise identical variant in which DMR is disabled. Both settings use the same checkpoint; no retraining is performed. Across five fidelity metrics, enabling DMR improves $\alpha$-precision by 1.7 % and yields consistent gains in the remaining scores (Trend +0.3 %, C2ST +0.6 %, and MLE +0.3 %). DMR therefore boosts not only $\alpha$-precision but overall fidelity.

**Label-leakage impact**   To examine the effect of label leakage, we retrained IMPUGEN after removing all label columns. Including labels increases MAE and RMSE by 0.4 % and 2.3 %, respectively, indicating that label information slightly degrades imputation accuracy instead of improving it.

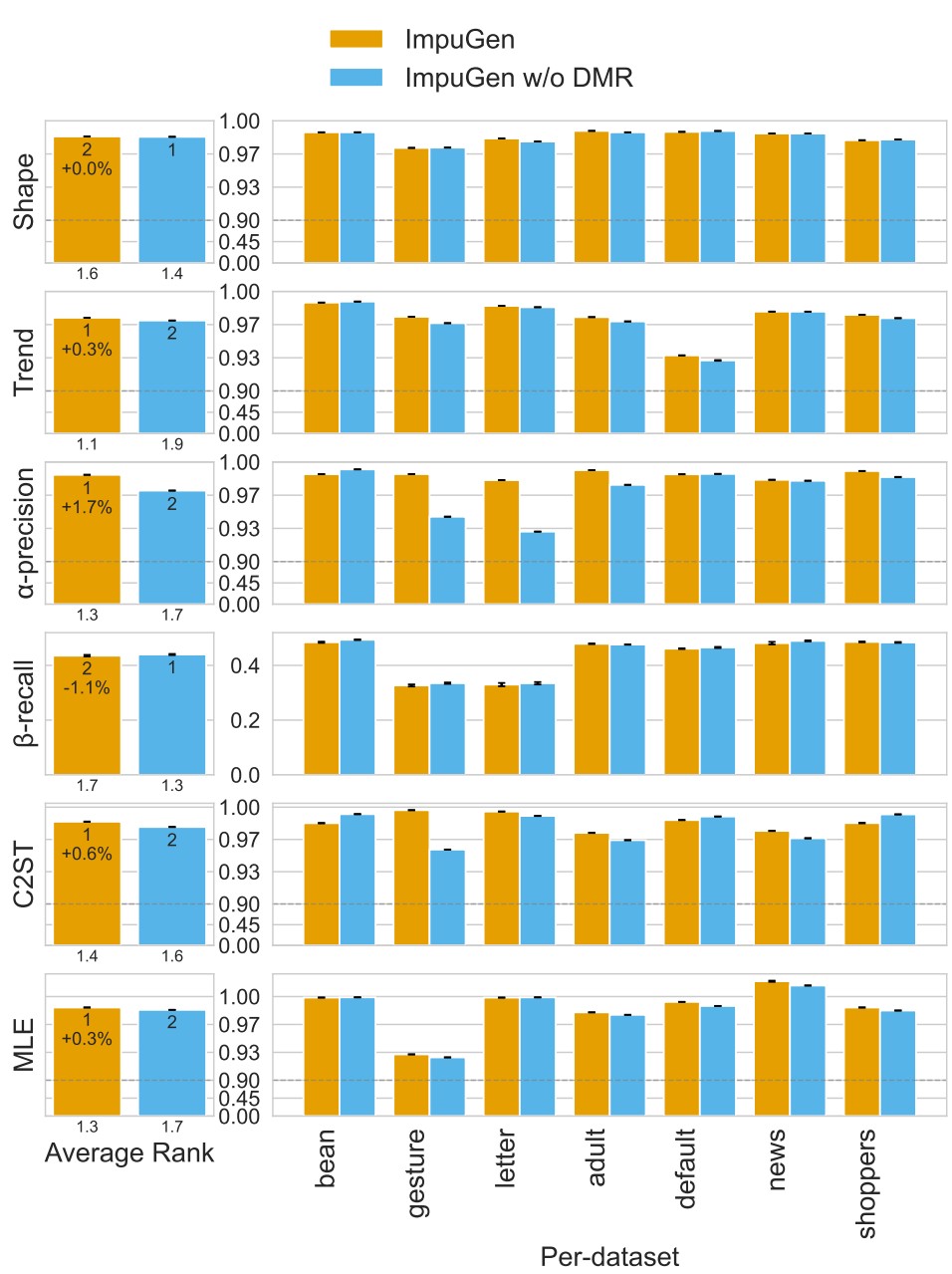

Figure 5: Effect of DMR sampling on synthetic-table fidelity. We evaluate IMPUGEN and an ablated variant without DMR on the seven data sets (Bean, Gesture, Letter, Adult, Default, News, Shoppers). **Left:** Mean score across data sets; the x-tick below each bar shows the corresponding average rank, and the value printed below indicates IMPUGEN's macro-average improvement over the ablation. **Right:** Per-data-set bars, grouped by metric (Shape, Trend, $\alpha$-precision, $\beta$-recall, C2ST, and MLE). A broken $y$-axis enlarges the 0.90–1.00 range to expose differences among saturated scores. On average, DMR raises $\alpha$-precision by 1.6 %, Trend by 0.4 %, C2ST by 0.5 %, and MLE by 0.2 %.

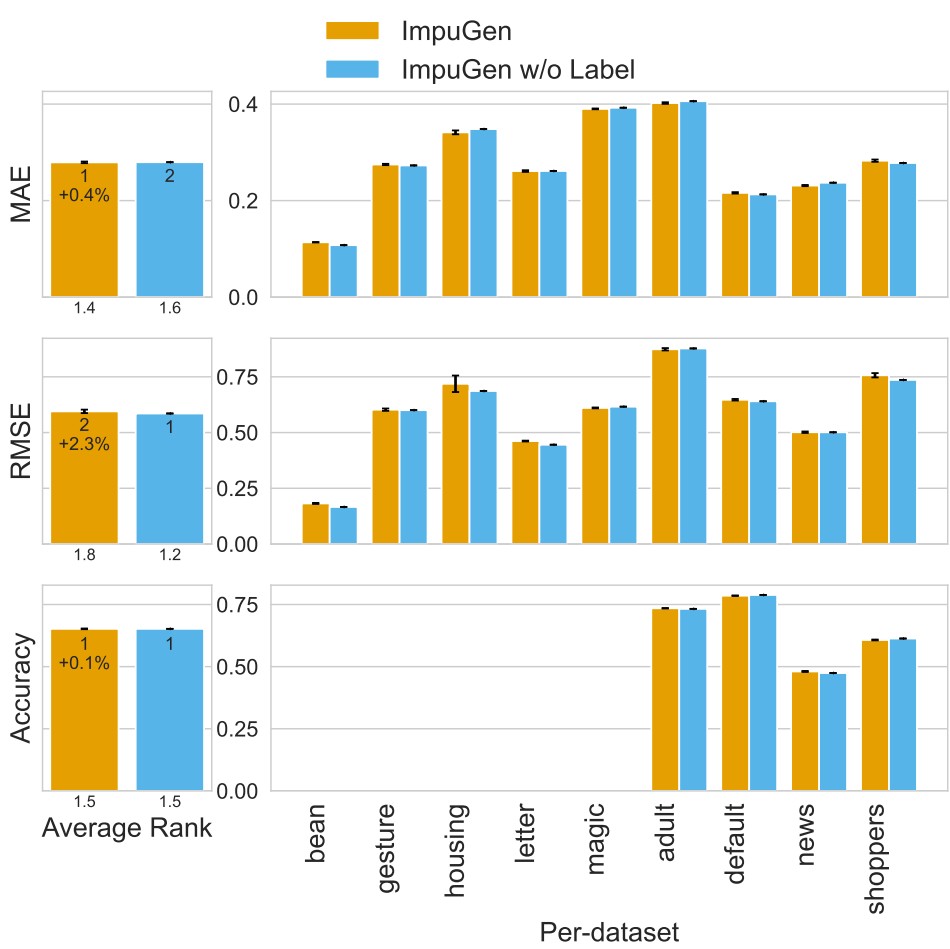

Figure 6: Effect of including labels during training. The standard (label-aware) IMPUGEN is compared with a variant trained without labels on nine data sets, addressing the concern that conditioning on labels might introduce target leakage. **Left.** Mean MAE, RMSE, and classification accuracy over all data sets; the x-tick below each bar shows the corresponding average rank. Percent values under the MAE and RMSE bars report the macro-average change relative to the label-agnostic model. **Right.** Per-data-set bars for the same metrics. The label-aware variant slightly increases MAE and RMSE and leaves accuracy essentially unchanged, suggesting that any leakage confers no benefit and can even hamper imputation.

## D    BASELINE IMPLEMENTATION DETAILS

All baselines are re-implemented from their official public repositories and wrapped in a `LightningModule` to ensure reproducible benchmarking. Unless stated otherwise, we keep the hyper-parameters reported by the original authors.

**Diffputer.**    Implementation follows the official code[1]. Ten posterior samples are drawn per data set and averaged. We run five EM iterations and apply the official `anabit` encoding to categorical columns; one-hot encoding did not reproduce the reported results.

**KnewImp.**    Implementation follows the official code[2]. Categorical features are one-hot encoded. To avoid out-of-memory errors during kernel computation, we introduce mini-batching with a batch size of 4,096, which increases RMSE on `Adult` by 2.5 %.

**SimpDM.**    Using the `power` configuration in the repository[3], we set the learning rate to $10^{-3}$, enable `ReduceLROnPlateau`, and apply early stopping. These changes reduce RMSE on `Adult` by 1.6 %.

**GRAPE.**    Implemented with default hyper-parameters from the official repository[4]

**ReMasker.**    The "Letter" configuration from the official repository[5] is used for every data set without modification.

**MaCoDE.**    We adopt the official implementation[6] and average ten imputations per missing entry, lowering RMSE on `Adult` by 17.5 %.

**HyperImpute, MissForest, EM, GAIN, and MICE.**    All methods are taken from the hyperimpute repository[7]. For MICE, we average ten imputations, which reduces RMSE on `Adult` by 34.9 %.

**TabNAT.**    Implemented with default hyper-parameters from the official repository[8]

**TabDiff.**    Implemented with default hyper-parameters from the official repository[9].

**TabSyn and TabDDPM.**    TabSyn follows the official implementation[10] with default settings. Although TabDDPM has its own repository, we use the unconditional tabddpm baseline bundled with TabSyn to ensure a consistent comparison framework.

**CTGAN and TVAE.**    Both models rely on the official CTGAN implementation[11]. Following the TabSyn (Zhang et al., 2024), we widen the generator and discriminator MLPs to match the layer widths used in their comparison experiments.

---

[1]https://github.com/hengruizhang98/DiffPuter
[2]https://github.com/JustusvLiebig/NewImp
[3]https://github.com/yixinliu233/SimpDM
[4]https://github.com/maxiaoba/GRAPE
[5]https://github.com/tydusky/remasker
[6]https://github.com/an-seunghwan/MaCoDE
[7]https://github.com/vanderschaarlab/hyperimpute
[8]https://github.com/fangliancheng/TabNAT
[9]https://github.com/MinkaiXu/TabDiff
[10]https://github.com/amazon-science/tabsyn
[11]https://github.com/sdv-dev/CTGAN

| Hyperparameter | Value |
| --- | --- |
| epochs | 10000 |
| early_stopping | 500 |
| batch_size | 4096 |
| lr | 0.0001 |
| optimizer | Adam |
| weight_decay | 0 |
| scheduler | ReduceLROnPlateau |
| em_steps | **5** |
| num_average | 10 |
| hid_dim | 1024 |
| categorical | **anabit** |
| continuous | standard |

Table 3: Diffputer hyper-parameters. Hyper-parameters different from the original implementation were shown as bold.

| Hyperparameter | Value |
| --- | --- |
| epochs | 200 |
| batch_size | 512 |
| lr | 0.1 |
| score_lr | 0.001 |
| optimizer | Adam |
| niter | 2 |
| mlp | [256, 256] |
| entropy_reg | 10 |
| bandwidth | 0.5 |
| noise | 0 |
| **kernel_batch_size** | **4096** |
| categorical | onehot |
| continuous | **quantile** |

Table 4: KnewImp hyper-parameters. Hyper-parameters different from the original implementation were shown as bold.

| Hyperparameter | Value |
| --- | --- |
| epochs | 10000 |
| **early_stopping** | **500** |
| batch_size | 4096 |
| lr | **0.001** |
| optimizer | Adam |
| weight_decay | 0 |
| scheduler | **ReduceLROnPlateau** |
| num_layers | 5 |
| hid_dim | 1024 |
| categorical | onehot |
| continuous | minmax |

Table 5: SimpDM hyper-parameters. Hyper-parameters different from the original implementation were shown as bold.

| Hyperparameter | Value |
|---|---|
| epochs | 600 |
| batch_size | 64 |
| lr | 0.001 |
| min_lr | 0.00001 |
| optimizer | Adam |
| weight_decay | 0 |
| scheduler | CosineLRWithWarmUp |
| num_layers | 5 |
| hid_dim | 1024 |
| **categorical** | **onehot** |
| continuous | minmax |

Table 6: ReMasker hyper-parameters. Hyper-parameters different from the original implementation were shown as bold.

| Hyperparameter | Value |
|---|---|
| epochs | 500 |
| batch_size | 1024 |
| lr | 0.001 |
| optimizer | Adam |
| weight_decay | 0.001 |
| **num_average** | **10** |
| d_transformer | 128 |
| num_transformer_heads | 4 |
| num_transformer_layer | 2 |
| tau | 1 |
| bins | 50 |
| categorical | LabelEncoder |
| continuous | – |

Table 7: MaCoDE hyper-parameters. Hyper-parameters different from the original implementation were shown as bold.

| Hyperparameter | Value |
|---|---|
| epochs | 5000 |
| batch_size | 1024 |
| lr | 0.001 |
| optimizer | Adam |
| weight_decay | 1e-6 |
| embed_dim | 32 |
| buffer_size | 8 |
| depth | 6 |
| categorical | LabelEncoder |
| continuous | quantile |

Table 8: TabNAT hyper-parameters.

| Hyperparameter | Value |
| --- | --- |
| epochs | 8000 |
| batch_size | 4096 |
| lr | 0.001 |
| optimizer | Adam |
| weight_decay | 0 |
| ema_decay | 0.997 |
| num_layers | 2 |
| d_token | 4 |
| n_head | 1 |
| factor | 32 |
| dim_t | 1024 |
| precond | TRUE |
| sigma_data | 1 |
| scheduler | power_mean_per_column |
| cat_scheduler | log_linear_per_column |
| noise_dist | uniform_t |
| categorical | LabelEncoder |
| continuous | quantile |

Table 9: TabDiff hyper-parameters

| Hyperparameter | Value |
| --- | --- |
| epochs | 10000 |
| early_stopping | 500 |
| batch_size | 4096 |
| lr | 0.001 |
| optimizer | Adam |
| weight_decay | 0 |
| scheduler | ReduceLROnPlateau |
| d_model | 1024 |
| categorical | LabelEncoder |
| continuous | quantile |

Table 10: TabSyn hyper-parameters

| Hyperparameter | Value |
| --- | --- |
| epochs | 4000 |
| batch_size | 4096 |
| lr | 0.001 |
| optimizer | Adam |
| weight_decay | 0 |
| scheduler | ReduceLROnPlateau |
| max_beta | 1e-2 |
| min_beta | 1e-5 |
| beta_decay | 0.7 |
| d_model | 1024 |
| num_layers | 2 |
| d_token | 4 |
| n_head | 1 |
| factor | 32 |
| categorical | LabelEncoder |
| continuous | quantile |

Table 11: TabSynVAE hyper-parameters

| Hyperparameter | Value |
| --- | --- |
| max_steps | 30000 |
| batch_size | 4096 |
| lr | 0.001809 |
| optimizer | AdamW |
| scheduler | LinearLR |
| diff_d_model | 1024 |
| **gradient_clip_val** | **0.5** |
| categorical | LabelEncoder |
| continuous | quantile |

Table 12: TabDDPM hyper-parameters

| Hyperparameter | Value |
| --- | --- |
| epochs | 5000 |
| batch_size | 500 |
| generator_lr | 0.0002 |
| discriminator_lr | 0.0002 |
| generator_decay | 0.000001 |
| discriminator_decay | 0.000001 |
| embedding_dim | 1024 |
| generator_dim | 1024,2048,2048,1024 |
| discriminator_dim | 1024,2048,2048,1024 |
| categorical | LabelEncoder |
| continuous | minmax |

Table 13: CTGAN hyper-parameters

| Hyperparameter | Value |
| --- | --- |
| epochs | 1000 |
| batch_size | 500 |
| l2scale | 0.00001 |
| loss_factor | 2 |
| generator_dim | 1024,2048,2048,1024 |
| discriminator_dim | 1024,2048,2048,1024 |
| categorical | LabelEncoder |
| continuous | minmax |

Table 14: TVAE hyper-parameters

Environment All experiments were run on a workstation with the following hardware and software configuration:

- **Operating system**: Windows 11
- **CPU**: AMD Ryzen 5 5600X
- **GPU**: NVIDIA GeForce RTX 3090 (24 GB VRAM) and NVIDIA GeForce RTX 5090 (32 GB VRAM)
- **Software**: CUDA 12.8, Python 3.12, PyTorch 2.8.0 (Paszke et al., 2019), and PyTorch-Lightning 2.5.2 (Falcon, 2019)

## E  LLM ASSISTANCE DISCLOSURE

We used a large language model–based assistant solely for language editing (grammar, wording, and readability) and discovery of potential references.

## F  ADDITIONAL IMPUTATION RESULTS

We follow the evaluation protocol of DIFFPUTER (Zhang et al., 2025a) and report both in-sample (train) and out-of-sample (test) imputation performance. Results are stratified by the missing-value rate.

**Baselines.** When the missingness is 30 %, every baseline is evaluated. For the in-sample setting this yields ten methods, while the out-of-sample setting is limited to six—Diffputer, KnewImp, SimpDM, ReMasker, MaCoDE, and MICE—because only these support out-of-sample prediction. At 50 % missingness and above, ReMasker and GAIN perform much worse than the other methods and are therefore excluded. Consequently, eight baselines are compared in-sample and five (out-of-sample) at the higher missingness levels.

Tables S15 and S16 report the macro-averaged percentage improvement of IMPUGEN over the strongest competitor on nine benchmark data sets. Across all scenarios, IMPUGEN achieves the highest accuracy.

| Mechanism | Rate (%) | ΔMAE | ΔRMSE | ΔAcc | Mechanism | Rate (%) | ΔMAE | ΔRMSE | ΔAcc |
|-----------|----------|------|-------|------|-----------|----------|------|-------|------|
| MCAR | 30 | 6.7 | 2.6 | 2.0 | MCAR | 30 | 6.7 | 3.7 | 1.8 |
| MCAR | 50 | 7.2 | 3.0 | 3.3 | MCAR | 50 | 7.8 | 5.5 | 3.4 |
| MCAR | 70 | 6.3 | 0.8 | 4.9 | MCAR | 70 | 7.4 | 4.6 | 4.9 |
| MAR | 30 | 9.2 | 3.0 | 2.0 | MAR | 30 | 12.0 | 7.2 | 2.3 |
| MAR | 50 | 8.2 | 3.8 | 3.2 | MAR | 50 | 10.3 | 6.5 | 3.5 |
| MAR | 70 | 9.0 | 4.3 | 4.6 | MAR | 70 | 9.8 | 6.1 | 4.7 |
| MNAR | 30 | 6.7 | 1.7 | 2.2 | MNAR | 30 | 11.7 | 8.3 | 2.3 |
| MNAR | 50 | 8.3 | 4.4 | 3.3 | MNAR | 50 | 9.5 | 7.2 | 3.6 |
| MNAR | 70 | 6.3 | 2.1 | 4.7 | MNAR | 70 | 7.2 | 4.0 | 4.9 |

Table 15: Relative **in-sample** macro improvement (%).

Table 16: Relative **out-of-sample** macro improvement (%).

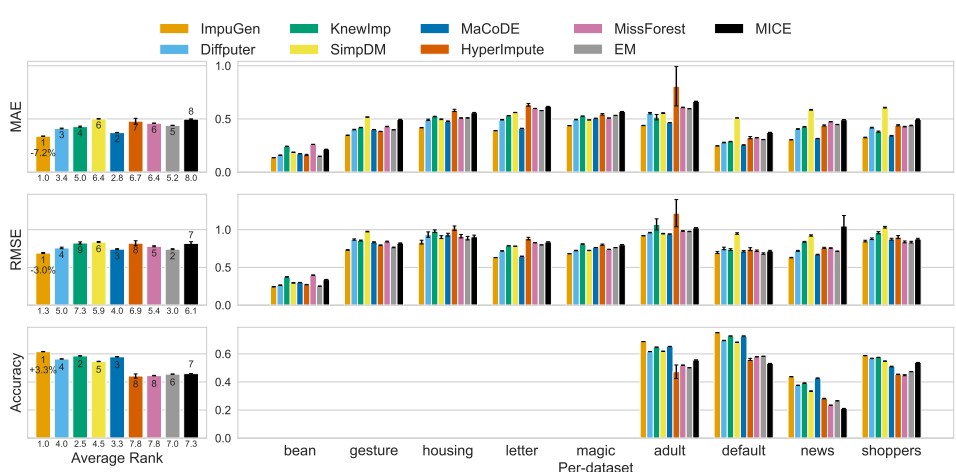

Figure 7: In-sample imputation results at 50 % MCAR missingness on nine data sets. IMPUGEN is compared with eight baselines on three metrics: MAE, RMSE, and categorical accuracy. ReMasker and GAIN are omitted because their performance drops sharply above 50 % missingness. The left panel shows the mean score and average rank for each metric; the percentage under the first bar indicates the average relative gain of IMPUGEN over the strongest baseline. The right panel reports per-data-set scores. On average, IMPUGEN reduces MAE by 7.2 % and RMSE by 3.0 % while achieving the highest accuracy rank.

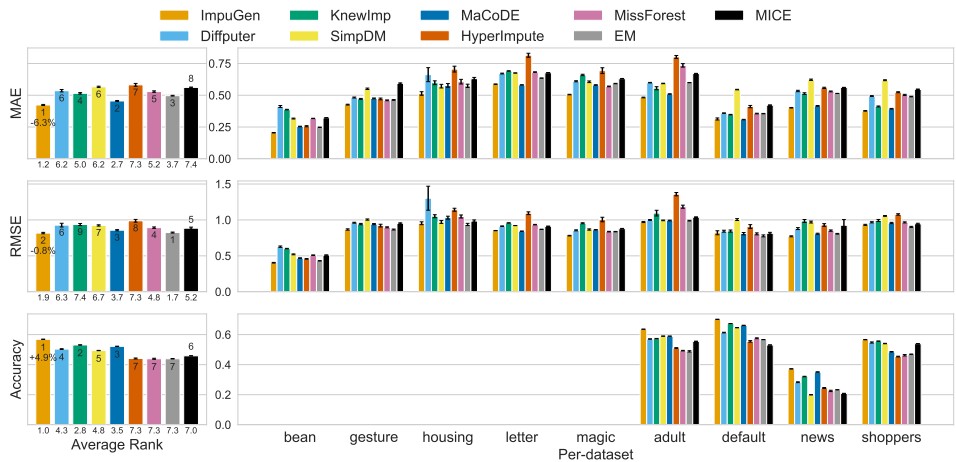

Figure 8: In-sample imputation results at 70 % MCAR missingness on nine data sets. IMPUGEN is evaluated against eight baselines on three metrics: MAE, RMSE, and categorical accuracy. Re-Masker and GAIN are omitted because their performance deteriorates above 50 % missingness. The left panel shows, for each metric, the mean score and average rank; the percentage beneath the first bar indicates IMPUGEN's average relative gain over the strongest competitor. The right panel displays per-data-set scores. On average, IMPUGEN lowers MAE by 6.3 % and RMSE by 0.8 % while maintaining the best accuracy rank.

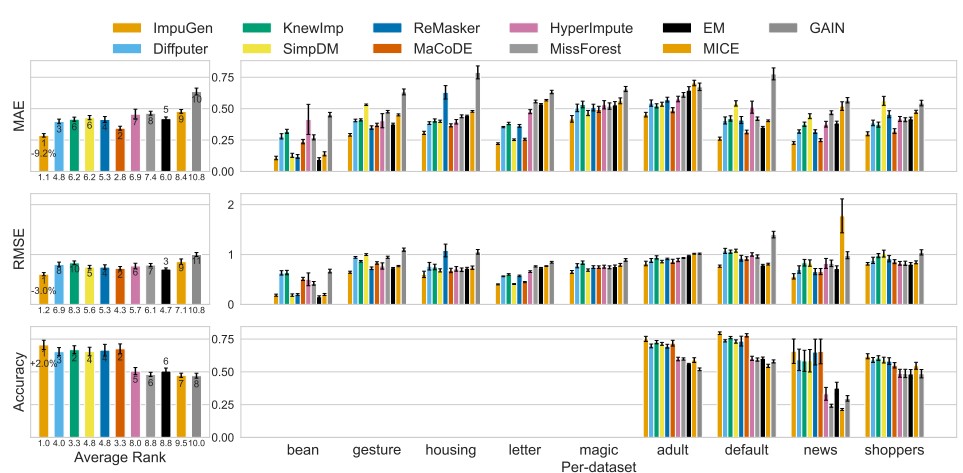

Figure 9: In-sample imputation results at 30 % MAR missingness on nine data sets. IMPUGEN is compared with ten baselines on three metrics: MAE, RMSE, and categorical accuracy. The left panel shows the mean score and average rank for each metric; the percentage under the first bar indicates the average relative gain of IMPUGEN over the strongest baseline. The right panel reports per-data-set scores. On average, IMPUGEN reduces MAE by 9.2 % and RMSE by 3.0 % while achieving the highest accuracy rank.

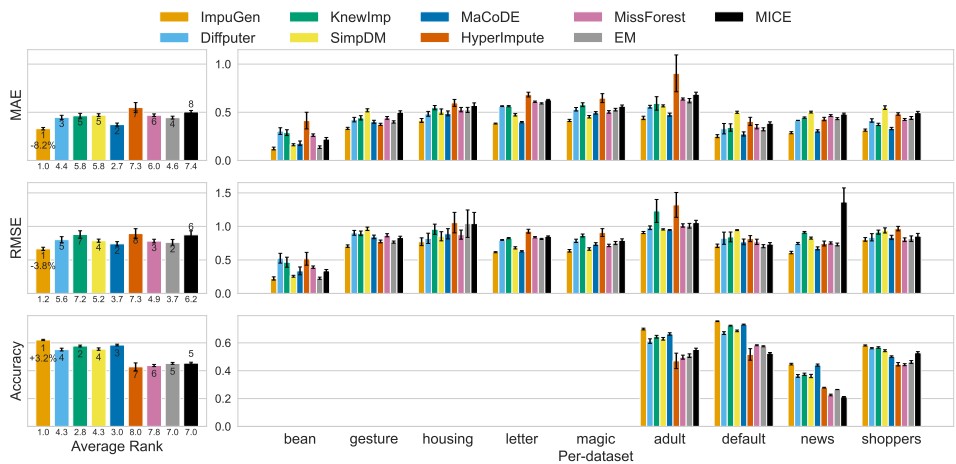

Figure 10: In-sample imputation results at 50 % MAR missingness on nine data sets. IMPUGEN is compared with eight baselines on three metrics: MAE, RMSE, and categorical accuracy. ReMasker and GAIN are omitted because their performance drops sharply above 50 % missingness. The left panel shows the mean score and average rank for each metric; the percentage under the first bar indicates the average relative gain of IMPUGEN over the strongest baseline. The right panel reports per-data-set scores. On average, IMPUGEN reduces MAE by 8.2 % and RMSE by 3.8 % while achieving the highest accuracy rank.

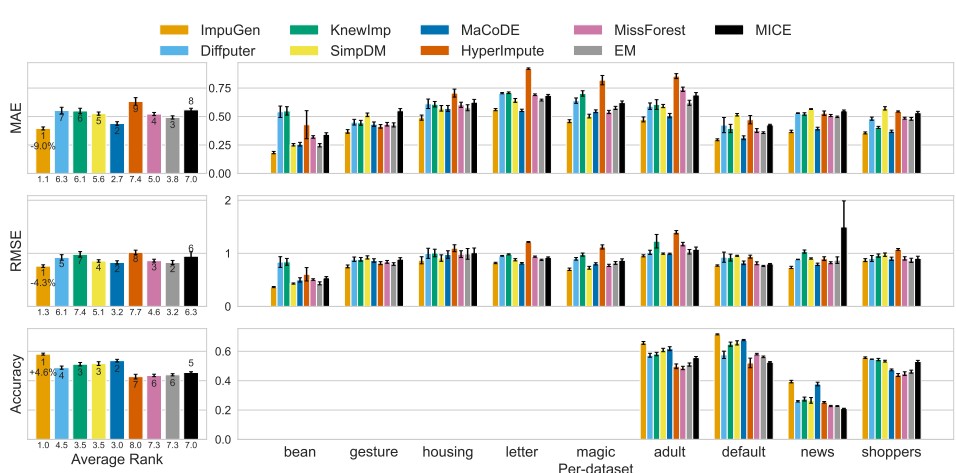

Figure 11: In-sample imputation results at 70 % MAR missingness on nine data sets. IMPUGEN is compared with eight baselines on three metrics: MAE, RMSE, and categorical accuracy. ReMasker and GAIN are omitted because their performance drops sharply above 50 % missingness. The left panel shows the mean score and average rank for each metric; the percentage under the first bar indicates the average relative gain of IMPUGEN over the strongest baseline. The right panel reports per-data-set scores. On average, IMPUGEN reduces MAE by 9.0 % and RMSE by 4.3 % while achieving the highest accuracy rank.

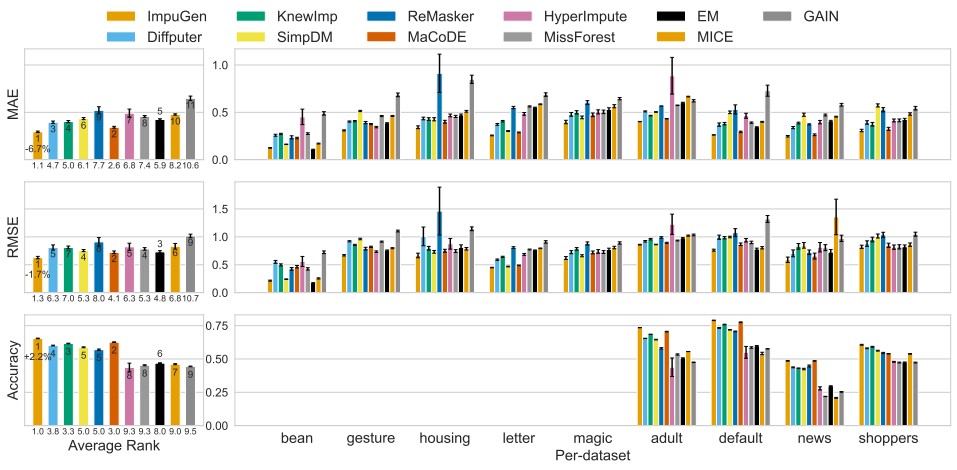

Figure 12: In-sample imputation results at 30 % MNAR missingness on nine data sets. IMPUGEN is compared with ten baselines on three metrics: MAE, RMSE, and categorical accuracy. The left panel shows the mean score and average rank for each metric; the percentage under the first bar indicates the average relative gain of IMPUGEN over the strongest baseline. The right panel reports per-data-set scores. On average, IMPUGEN reduces MAE by 6.7 % and RMSE by 1.7 % while achieving the highest accuracy rank.

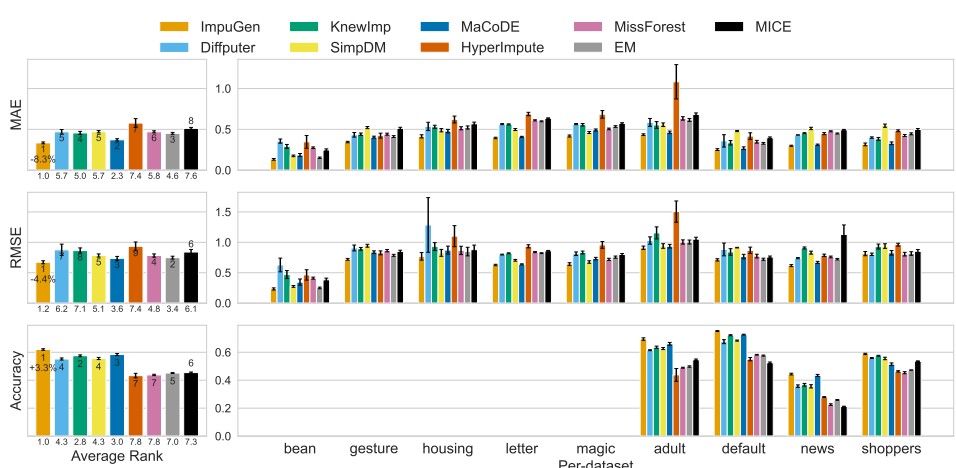

Figure 13: In-sample imputation results at 50 % MNAR missingness on nine data sets. IMPUGEN is compared with eight baselines on three metrics: MAE, RMSE, and categorical accuracy. ReMasker and GAIN are omitted because their performance drops sharply above 50 % missingness. The left panel shows the mean score and average rank for each metric; the percentage under the first bar indicates the average relative gain of IMPUGEN over the strongest baseline. The right panel reports per-data-set scores. On average, IMPUGEN reduces MAE by 8.3 % and RMSE by 4.4 % while achieving the highest accuracy rank.

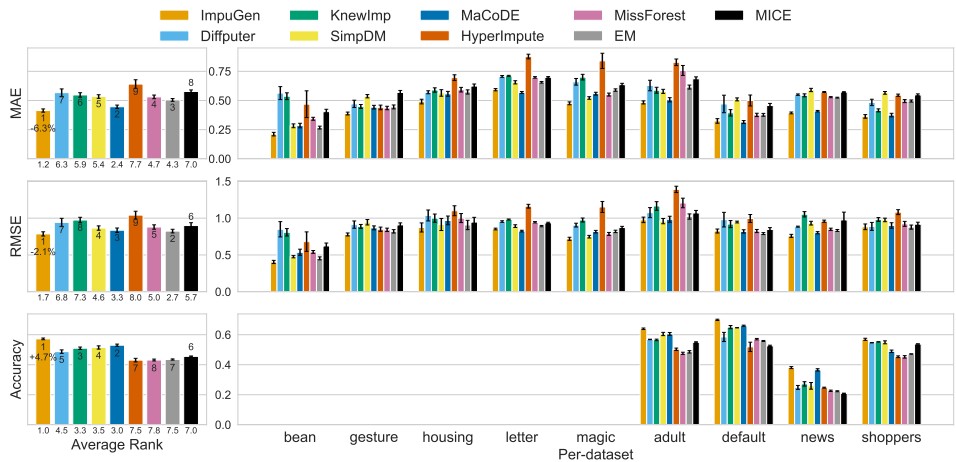

Figure 14: In-sample imputation results at 70 % MNAR missingness on nine data sets. IMPUGEN is compared with eight baselines on three metrics: MAE, RMSE, and categorical accuracy. ReMasker and GAIN are omitted because their performance drops sharply above 50 % missingness. The left panel shows the mean score and average rank for each metric; the percentage under the first bar indicates the average relative gain of IMPUGEN over the strongest baseline. The right panel reports per-data-set scores. On average, IMPUGEN reduces MAE by 6.3 % and RMSE by 2.1 % while achieving the highest accuracy rank.

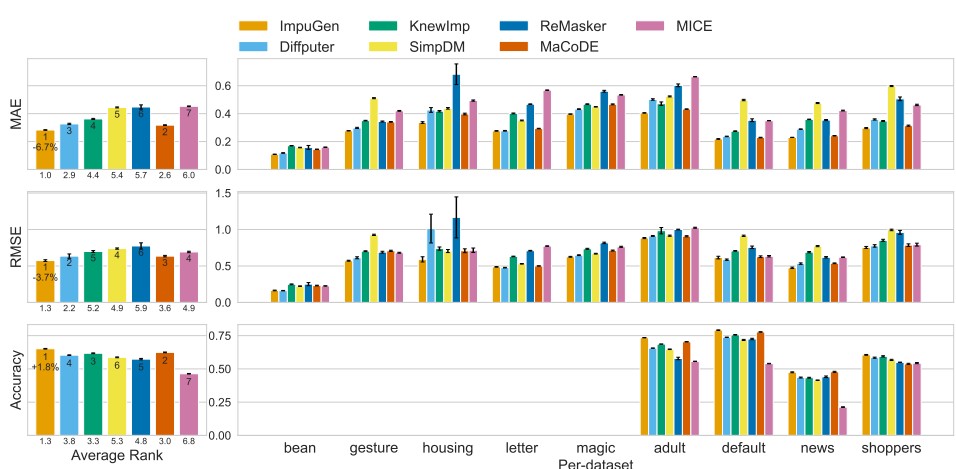

Figure 15: Out-of-sample imputation results at 30 % MCAR missingness on nine data sets. IMPU-GEN is compared with six baselines on three metrics: MAE, RMSE, and categorical accuracy. The left panel shows the mean score and average rank for each metric; the percentage under the first bar indicates the average relative gain of IMPUGEN over the strongest baseline. The right panel reports per-data-set scores. On average, IMPUGEN reduces MAE by 6.7 % and RMSE by 3.7 % while achieving the highest accuracy rank.

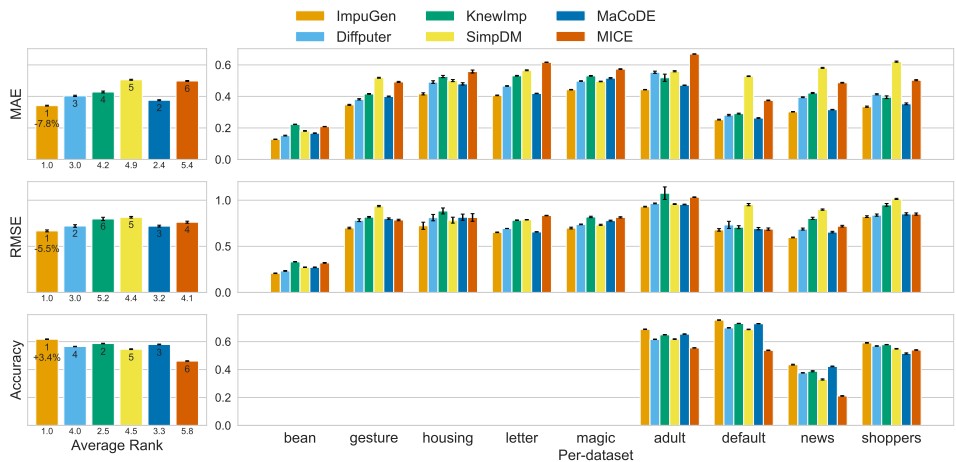

Figure 16: Out-of-sample imputation results at 50 % MCAR missingness on nine data sets. IM-PUGEN is compared with five baselines on three metrics: MAE, RMSE, and categorical accuracy. ReMasker is omitted because its performance drops sharply above 50 % missingness. The left panel shows the mean score and average rank for each metric; the percentage under the first bar indicates the average relative gain of IMPUGEN over the strongest baseline. The right panel reports per-data-set scores. On average, IMPUGEN reduces MAE by 7.8 % and RMSE by 5.5 % while achieving the highest accuracy rank.

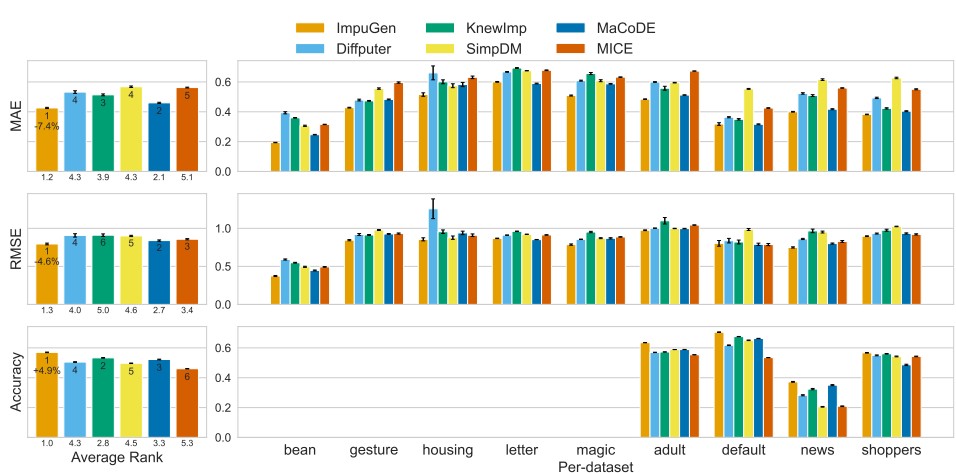

Figure 17: Out-of-sample imputation results at 70 % MCAR missingness on nine data sets. IM-PUGEN is compared with five baselines on three metrics: MAE, RMSE, and categorical accuracy. ReMasker is omitted because its performance drops sharply above 50 % missingness. The left panel shows the mean score and average rank for each metric; the percentage under the first bar indicates the average relative gain of IMPUGEN over the strongest baseline. The right panel reports per-data-set scores. On average, IMPUGEN reduces MAE by 7.4 % and RMSE by 4.6 % while achieving the highest accuracy rank.

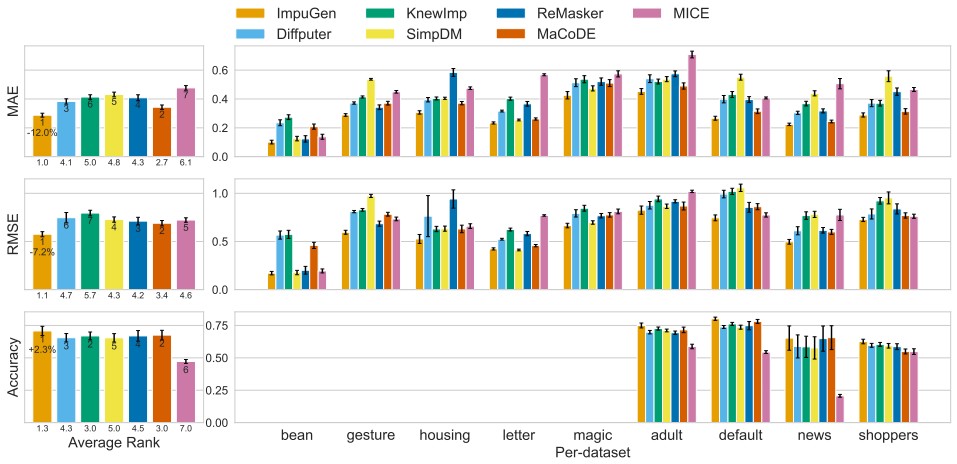

Figure 18: Out-of-sample imputation results at 30 % MAR missingness on nine data sets. IMPUGEN is compared with six baselines on three metrics: MAE, RMSE, and categorical accuracy. The left panel shows the mean score and average rank for each metric; the percentage under the first bar indicates the average relative gain of IMPUGEN over the strongest baseline. The right panel reports per-data-set scores. On average, IMPUGEN reduces MAE by 12.0 % and RMSE by 7.2 % while achieving the highest accuracy rank.

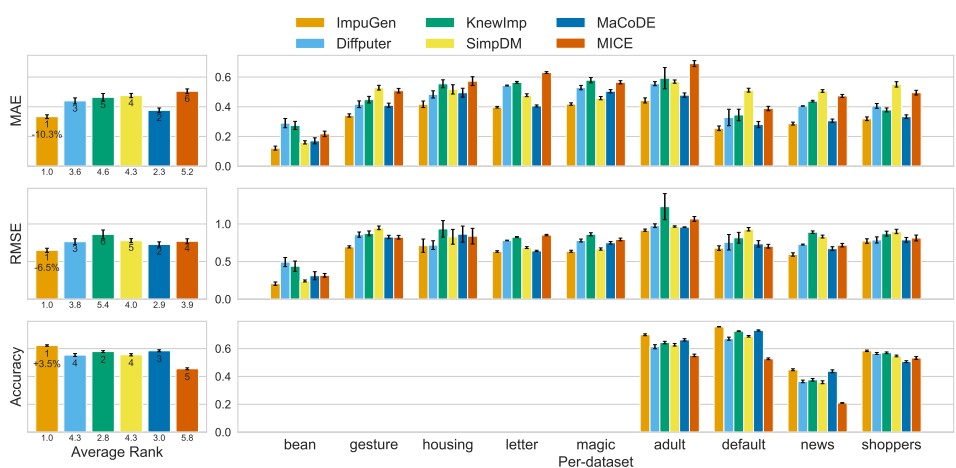

Figure 19: Out-of-sample imputation results at 50 % MAR missingness on nine data sets. IMPUGEN is compared with five baselines on three metrics: MAE, RMSE, and categorical accuracy. ReMasker is omitted because its performance drops sharply above 50 % missingness. The left panel shows the mean score and average rank for each metric; the percentage under the first bar indicates the average relative gain of IMPUGEN over the strongest baseline. The right panel reports per-data-set scores. On average, IMPUGEN reduces MAE by 10.3 % and RMSE by 6.5 % while achieving the highest accuracy rank.

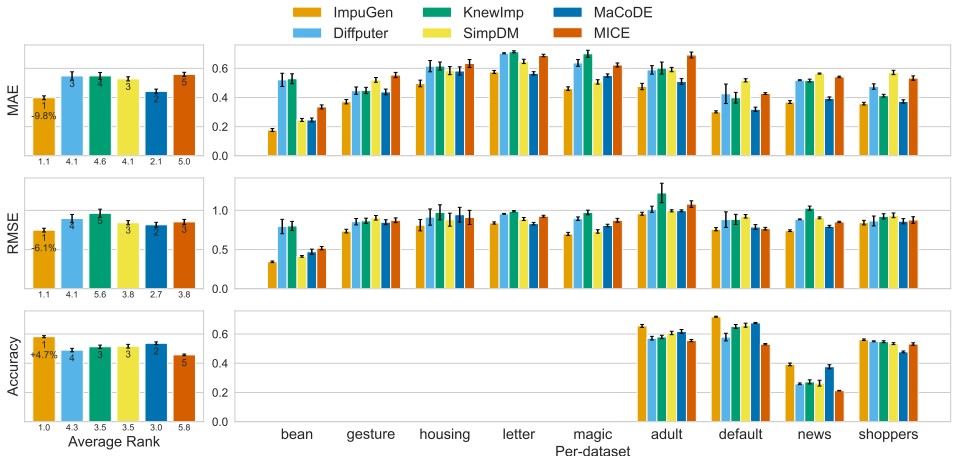

Figure 20: Out-of-sample imputation results at 70 % MAR missingness on nine data sets. IMPUGEN is compared with five baselines on three metrics: MAE, RMSE, and categorical accuracy. ReMasker is omitted because its performance drops sharply above 50 % missingness. The left panel shows the mean score and average rank for each metric; the percentage under the first bar indicates the average relative gain of IMPUGEN over the strongest baseline. The right panel reports per-data-set scores. On average, IMPUGEN reduces MAE by 9.8 % and RMSE by 6.1 % while achieving the highest accuracy rank.

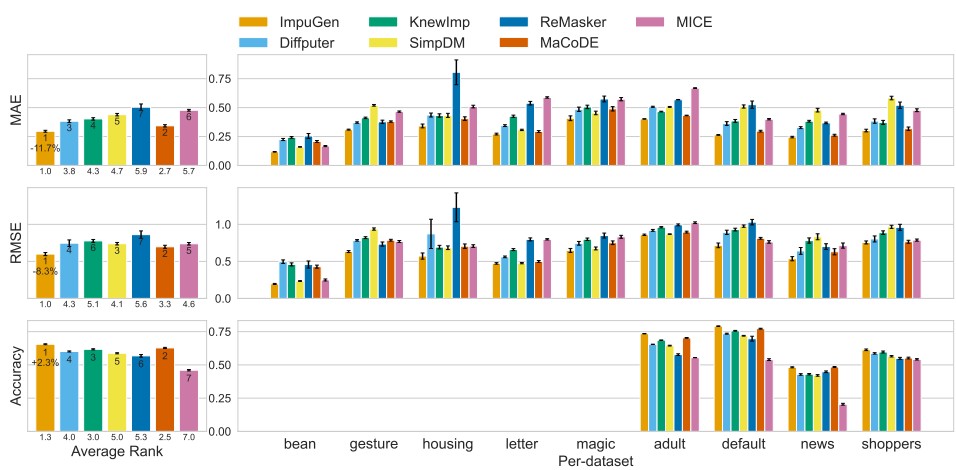

Figure 21: Out-of-sample imputation results at 30 % MNAR missingness on nine data sets. IMPU-GEN is compared with six baselines on three metrics: MAE, RMSE, and categorical accuracy. The left panel shows the mean score and average rank for each metric; the percentage under the first bar indicates the average relative gain of IMPUGEN over the strongest baseline. The right panel reports per-data-set scores. On average, IMPUGEN reduces MAE by 11.7 % and RMSE by 8.3 % while achieving the highest accuracy rank.

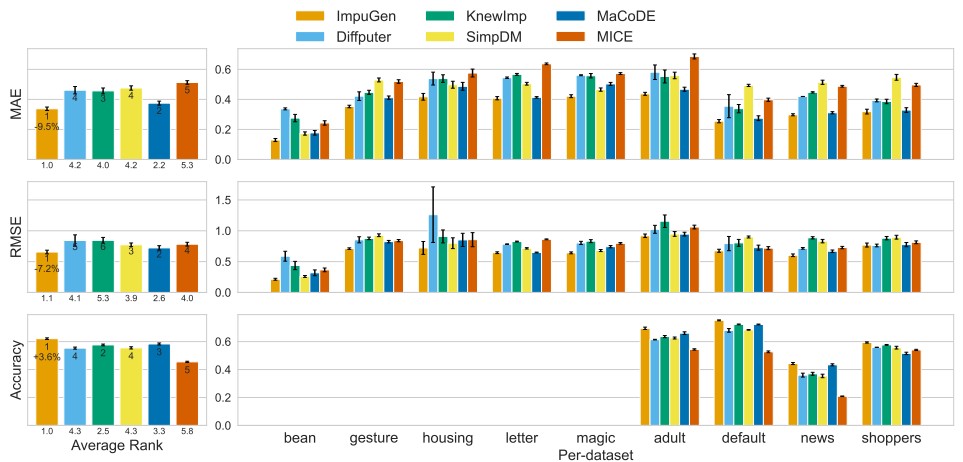

Figure 22: Out-of-sample imputation results at 50 % MNAR missingness on nine data sets. IM-PUGEN is compared with five baselines on three metrics: MAE, RMSE, and categorical accuracy. ReMasker is omitted because its performance drops sharply above 50 % missingness. The left panel shows the mean score and average rank for each metric; the percentage under the first bar indicates the average relative gain of IMPUGEN over the strongest baseline. The right panel reports per-data-set scores. On average, IMPUGEN reduces MAE by 9.5 % and RMSE by 7.2 % while achieving the highest accuracy rank.

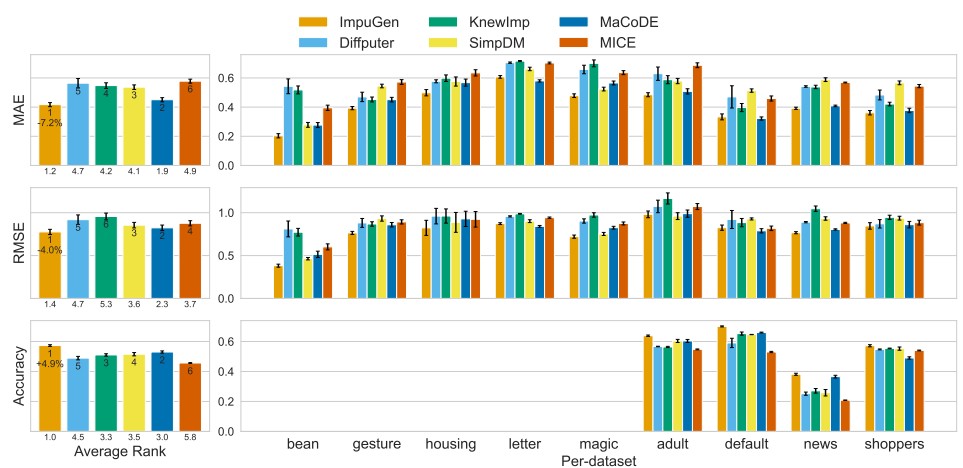

Figure 23: Out-of-sample imputation results at 70 % MNAR missingness on nine data sets. IM-PUGEN is compared with five baselines on three metrics: MAE, RMSE, and categorical accuracy. ReMasker is omitted because its performance drops sharply above 50 % missingness. The left panel shows the mean score and average rank for each metric; the percentage under the first bar indicates the average relative gain of IMPUGEN over the strongest baseline. The right panel reports per-dataset scores. On average, IMPUGEN reduces MAE by 7.2 % and RMSE by 4.0 % while achieving the highest accuracy rank.

