# OpenReview forum: "ImpuGen: Unified Tabular Imputation and Generation via Task-Aligned Sampling Strategies"
_ICLR.cc/2026/Conference — ICLR 2026 Conference Withdrawn Submission_

### Official Review · Reviewer_Fhxi · 2025-10-30

**Soundness:** 2
**Presentation:** 1
**Contribution:** 1
**Rating:** 2
**Confidence:** 5

**Summary:**

This paper introduces ImpuGen, which aims to unify tabular imputation and unconditional generation under a single diffusion-based framework. The core contribution is two sampling strategies for imputation and unconditional generation, respectively.

**Strengths:**

1. The idea of unifying imputation and unconditional generation under a single probabilistic framework is conceptually appealing.

2.  The proposed two sampling methods (for generation and imputation) are conceptually simple and seem interesting.

3.  The proposed method achieves the best performance compared to baselines.

**Weaknesses:**

1. **Poor writing and organization**.
The writing of this paper requires major revision.  Critical technical components are introduced without proper definition or citation, to name a few examples:

- In line 147, “AdaLN-Zero MLP” is mentioned but neither defined nor referenced anywhere in the text.

- In line 150, the important baseline MissDiff is cited in text but missing from the references.

- In line 134, the citation to EDM is missing, making the definition of the diffusion process incomplete.

Such omissions suggest a lack of proofreading and make the technical content extremely difficult to follow.
Moreover, the structure of the paper is disorganized: the problem definition and the EDM diffusion can be moved to the preliminary section instead of the main method section.

2. **Missing critical content and incomplete exposition**
Key theoretical and methodological components are missing from the paper. Notably, both the statement and proof of Theorem A (line 175) are absent, even though this theorem seems central to explaining the proposed zero-start sampling strategy. Without the theorem, the reader cannot verify the correctness, intuition, or derivation of the sampling approach.

Overall, even if the experimental results show promising performance, the writing quality and lack of technical clarity make it impossible to verify the validity of these results or to understand how the proposed method achieves such improvements. So I recommend rejection of the paper.

**Questions:**

See weakness.

---

### Official Review · Reviewer_rhSd · 2025-10-30

**Soundness:** 2
**Presentation:** 3
**Contribution:** 3
**Rating:** 4
**Confidence:** 4

**Summary:**

This paper proposes ImpuGen, a unified conditional diffusion model for both tabular data imputation and generation. The key contributions are two task-aligned sampling strategies: (1) zero-start sampling, which initializes the reverse diffusion process at 0 to produce deterministic imputations without ensemble averaging, and (2) distribution-matching refinement (DMR), which randomly re-masks columns with probability p and regenerates them. Experiments on nine benchmark datasets demonstrate that ImpuGen outperforms 11 imputation baselines and is competitive with state-of-the-art tabular generation approaches. Overall, this is solid empirical work with practical contributions, but the evaluations can be meaningfully strengthened.

**Strengths:**

1. Zero-start sampling is an elegant solution to achieve deterministic imputation point estimates.
The authors justify this choice by showing convergence to the conditional median in the 1D case

2. Table 2 demonstrates that ImpuGen achieves comparable or better accuracy than DiffPuter while being >100x faster.

3. The manuscript is also well-written and is easy for the reader to follow.

**Weaknesses:**

1. While DMR empirically improves results, the paper lacks clear intuition for why random re-masking helps. The motivation states it "nudges the sample distribution toward the empirical one" but doesn't explain the mechanism. Does it function as a form of data augmentation? Or something else? Recent works in imputation [1] have shown that random masks are at best optimal for the MCAR setting so the DMR is only expected it improve performance under MCAR (with feature independence) only. Additionally, the benchmark against this [1] method is missing. Also, ReMasker and GAIN should be included across all settings and missingness rates but are missing from the appendix results.

2. While ImpuGen achieves on average top-ranked performance, the actual improvements for imputation are sometimes small (are they statistically meaningful?). Or there’s a lack of improvement for quite a few generation metrics, like beta-recall and DCR.

3. The 1D theoretical analysis does not clearly extend to high-dimensional tabular data with mixed types (and correlated columns). The claim that zero-start converges to the conditional median needs stronger support. How does variable context-conditioning on observed entries affect this property?

#### References
[1] CACTI: Leveraging Copy Masking and Contextual Information to Improve Tabular Data Imputation (ICML 2025)

**Questions:**

1. The main contributions are sampling strategies rather than modeling innovations.
The backbone is a standard deterministic EDM with AdaLN-Zero conditioning. The paper would benefit from discussing why this particular architecture was chosen over alternatives.

2. The label leakage analysis shows that including labels during training slightly degrades imputation performance, which is counterintuitive. Can the authors explain the intuition for this further? Is there some unfair advantage ImpuGen has over pure imputation baselines?

3. How does performance scale with dataset size and dimensionality? All datasets are relatively small (<50K rows, <50 columns). How does the method perform on larger-scale problems?

4. Have the authors experimented with other deterministic initializations (e.g., mean imputation, linear interpolation from observed values)?

5. Figure 1 quality could be improved (small fonts, dense visualizations).

---

### Official Review · Reviewer_gzve · 2025-11-03

**Soundness:** 4
**Presentation:** 4
**Contribution:** 4
**Rating:** 8
**Confidence:** 4

**Summary:**

Use Adaptive layer norm to feed the data with missing values, but otherwise learn to denoise the whole data with diffusion. This allows conditioning only on non-missing data and generating the imputation. They start from a zero vector making it deterministic and starting from the mode of the diffusion. From this imputation, they use remasking and denoising to help improve quality further (similar to repaint).

Init at zero is novel for diffusion and it reminds me of the BigGAN trick. Its good because its deterministic which aligns it with MissForest and the like.

Overall, the idea is clear from the figure and makes a lot of sense.

The fact that you beat MissForest and MICE is incredible, because these are extremely strong baselines.

MCAR 30% is great to have, this is very aligned with the real world and results are good.

The experiment section is extremely well made and detailed.

**Strengths:**

See summary

**Weaknesses:**

The sketch proof A.1 is very limited, I have an hard time being convinced that it converges to the median. Normally L2 leads to mean and L1 leads to median. This theory part needs some work in my opinion. Its better left without theory than giving possibly an incorrect answer.

**Questions:**

Shouldnt it converge to the average if you use a L2 loss actually instead of the median or the mode? The sketch proof didnt convince me.

---

### Note · Authors · 2025-11-25

I have read and agree with the venue's withdrawal policy on behalf of myself and my co-authors.